# Epicardial VEGFC/D signaling is essential for coronary lymphangiogenesis

Ester de la Cruz[1], Vanessa Cadenas[1,2], Susana Temiño[1,2], Guillermo Oliver [ID][3] & Miguel Torres [ID][1,2✉]

## Abstract

The contractile ability of the mammalian heart critically relies on blood coronary circulation, essential to provide oxygen and nutrients to myocardial cells. In addition, the lymphatic vasculature is essential for the myocardial immune response, extracellular fluid homeostasis and response to injury. Recent studies identified different origins of coronary lymphatic endothelial cells, however, the cues that govern coronary lymphangiogenesis remain unknown. Here we show that the coronary lymphatic vasculature develops in intimate contact with the epicardium and with epicardial-derived cells. The epicardium expresses the lymphangiogenic cytokine VEGFC and its conditional deletion in the epicardium abrogates coronary lymphatic vasculature development. Interestingly, VEGFD is also expressed in the epicardium and cooperates with VEGFC in coronary lymphangiogenesis, but it does so only in females, uncovering an unsuspected sex-specific role for this cytokine. These results identify the epicardium/subepicardium as a signaling niche required for coronary lymphangiogenesis and VEGFC/D as essential mediators of this role.

Keywords Vascular Biology; Lymphatic Vasculature; Heart Development; Coronary Vasculature; Sex-specific Gene Function
Subject Categories Development; Vascular Biology & Angiogenesis

## Introduction

Besides the blood coronary vasculature, the ventricular myocardium also contains a fully developed network of cardiac lymphatics (Brakenhielm and Alitalo, 2019). Coronary lymphatics promote cardiomyocyte proliferation (Liu et al, 2020) and are important in the response to myocardial infarction (Henri et al, 2016; Klotz et al, 2015; Liu et al, 2020; Maruyama et al, 2021; Vieira et al, 2018) congestive heart failure (Witte et al, 1969), atherosclerosis (Lim et al, 2013; Milasan et al, 2016) and heart regeneration in zebrafish (Gancz et al, 2019; Harrison et al, 2019; Vivien et al, 2019). Cardiac lymphatics are first seen along the great arteries and sinus venosus at E12.5-E13.5 (Flaht et al, 2012; Karunamuni et al, 2010). Most embryonic and cardiac lymphatic endothelial cells (LECs) have a venous origin (Klotz et al, 2015; Wigle and Oliver, 1999); however, mesenchymal precursors from the second heart field (SHF) also contribute to ventral coronary LECs (Lioux et al, 2020; Maruyama et al, 2019). The transcription factor *Prox1* (Wigle and Oliver, 1999), as well as the growth factors VEGFC and VEGFD and their receptor VEGFR3 (Baldwin et al, 2005; Bower et al, 2017; Haiko et al, 2008; Karkkainen et al, 2004; Karkkainen et al, 2001; Paquet-Fifield et al, 2013) are essential during developmental lymphangiogenesis. In the heart, local signaling pathways like Retinoic acid (RA) and macrophage-derived hyaluronan promote cardiac lymphatic vessel maturation and remodeling (Cahill et al, 2021; Lioux et al, 2020), while the Sema3E-PlexinD1 axis is involved in coronary artery and lymphatic vessel patterning (Maruyama et al, 2021). Despite this progress, the cellular and molecular drivers of coronary lymphangiogenesis in the developing heart remain unknown. In some tissues/organs, arteries guide collector lymphatic vessel growth through VEGFC and CXCL12 expression from endothelial and smooth muscle cells (Cha et al, 2012; Vaahtomeri et al, 2017). Here, we characterize the growth pattern of coronary lymphatic vessels and report that they do not follow arteries or veins as they colonize the ventricles but, instead, rely on intimate interaction with the epicardium and epicardial-derived cells. Functional genetic analyses demonstrate that VEGFC/D are essential for the lymphangiogenic function of the epicardium.

## Results and discussion

To identify the mechanisms that guide coronary lymphangiogenesis, we studied the distribution of cardiac lymphatics as they grow into the ventricles of the mouse fetal heart (Fig. 1A–C"). Cardiac lymphatics appear at the base of the ventricles, dorsally at the sinus venosus and ventrally on the great arteries around embryonic day 12.5 (E12.5), however, they do not start colonizing the ventricles until E14.5 (reviewed in (Klaourakis et al, 2021). At E16.5, coronary veins and arteries are already present; however, lymphatic vessels do not follow the pattern of coronary arteries or veins as they colonize the ventricles (Fig. 1A–B",D). In histological sections at E15.5 we observed that coronary lymphatic vessels grow in direct association with the epicardium (Fig. 1C–C"). Coronary veins also grow within the subepicardial space, however, when lymphatic vessels coincide with veins in the subepicardium, the lymphatic

[1]Cardiovascular Regeneration Program, Centro Nacional de Investigaciones Cardiovasculares (CNIC), Madrid 28029, Spain. [2]Centro de Investigación Biomédica en Red de Enfermedades Cardiovasculares (CIBERCV), Madrid, Spain. [3]Center for Vascular and Developmental Biology, Feinberg Cardiovascular and Renal Research Institute, Northwestern University, Chicago, IL 60611, USA. ✉E-mail: mtorres@cnic.es

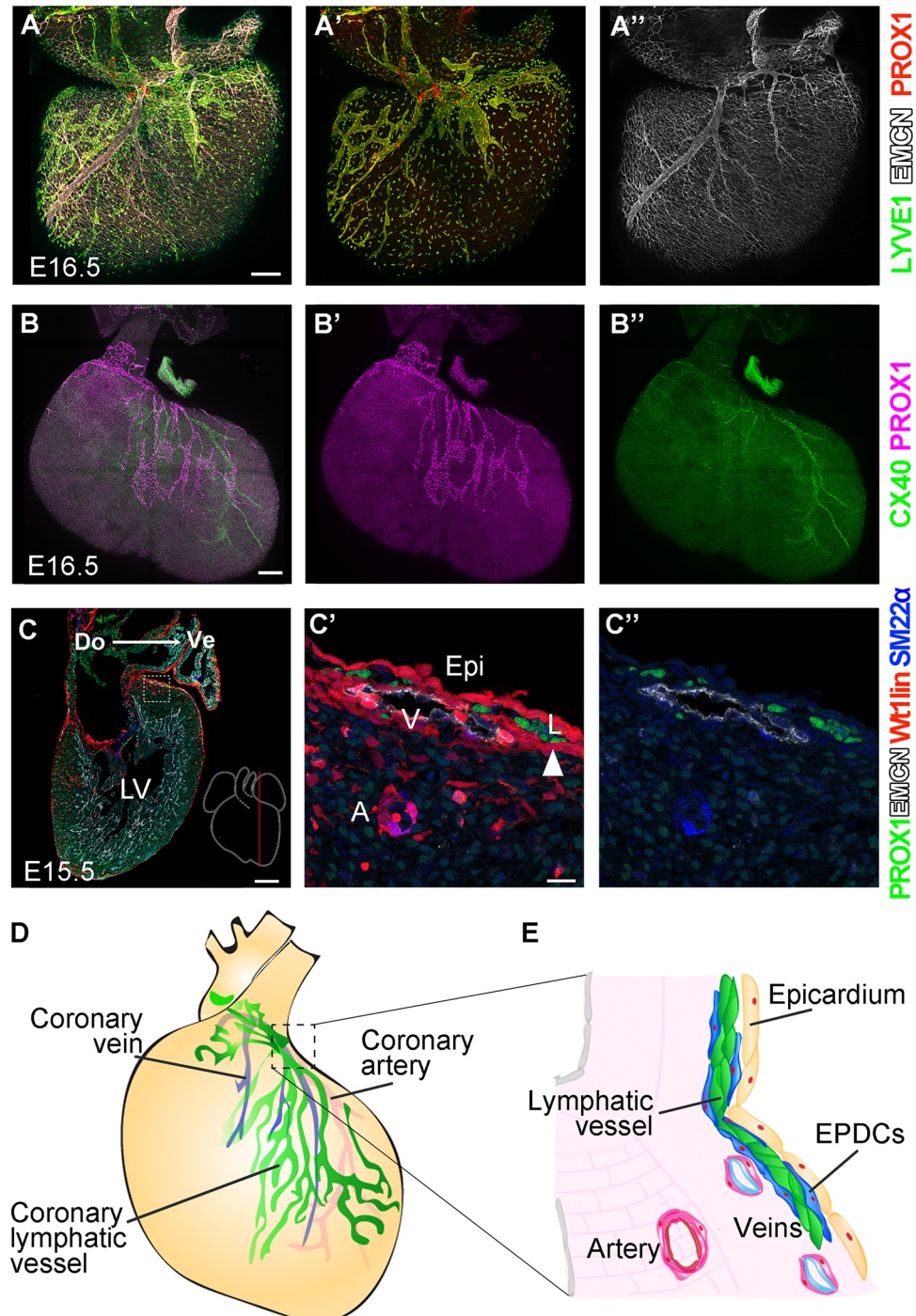

**Figure 1. Subepicardial coronary lymphangiogenesis.**

(A–A") Detection of veins with Endomucin (EMCN) and lymphatic vessels with Lyve1 and Prox1 in the dorsal side of E16.5 mouse hearts. (B–B"). Detection of arteries with Connexin 40 (CX40) and lymphatic vessels with PROX1 in the ventral side of an E16.5 mouse heart. (C–C") Detection of arteries (A), veins (V), and lymphatic vessels (L) in a sagittal section of an E15.5 Wt1$^{Cre}$; Rosa26R$^{Tomato}$ mouse heart, which reveals the lineage of cells that express Wt1$^{Cre}$ (Wt1lin), including the epicardium (Epi). Arrowhead in (C') indicates lymphatic vessel-associated EPDCs. Do: dorsal; Ve: ventral; LV, left ventricle. (D, E) Schemes of the disposition of coronary vessels in a developing mouse heart. EPDC: epicardial-derived cells. Scale bars 200 μm in (A), (B), (C) and 20 μm in (C').

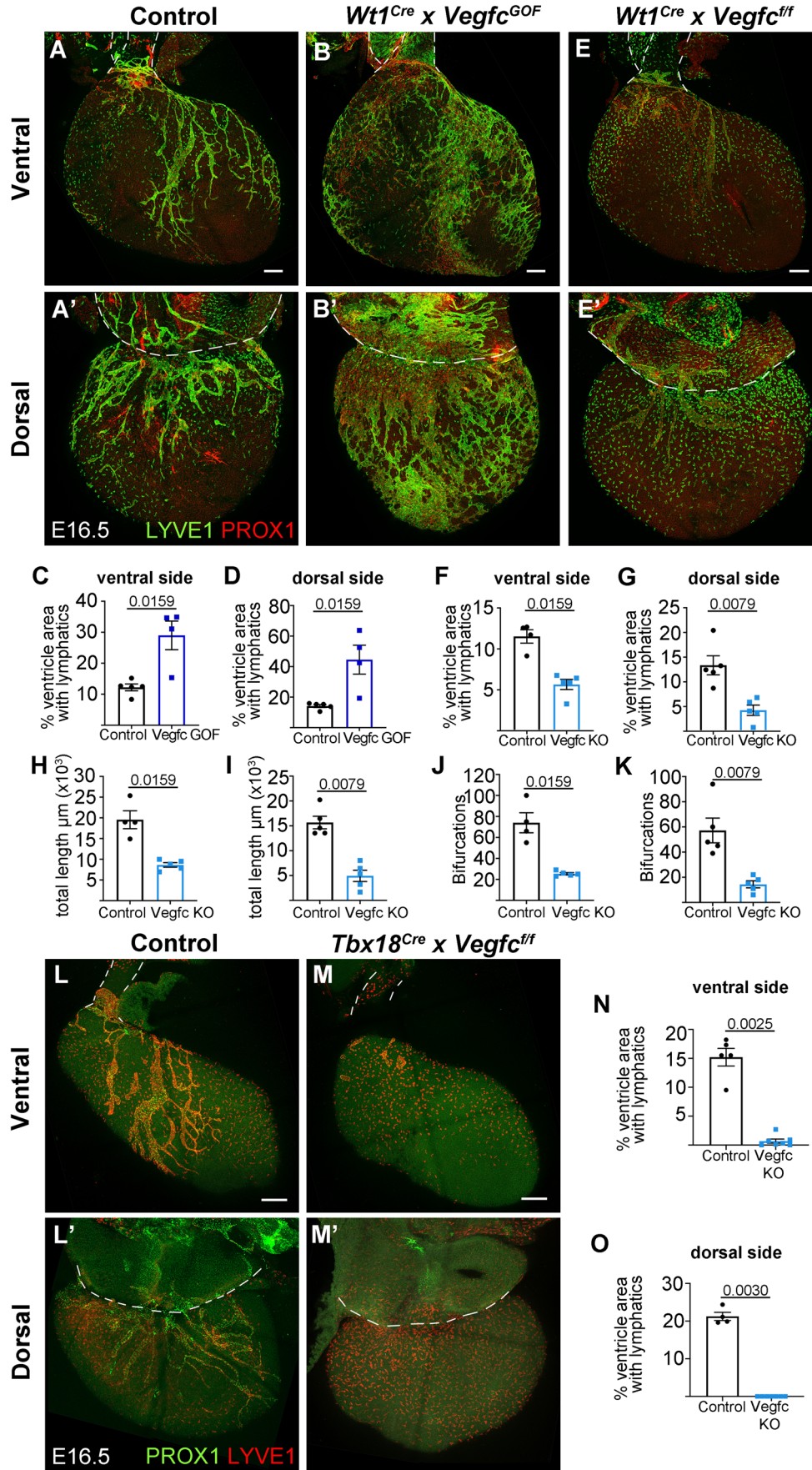

**Figure 2. Epicardial VEGFC is essential for coronary lymphangiogenesis.**

(A–D) Representative specimens and quantifications of lymphatic vessel coverage of the dorsal and ventral ventricular surfaces upon *Vegfc* overexpression by activation of *Eef1a1^VegfcGOF* with *Wt1^Cre* (*Vegfc ^GOF*). (E–K) Representative specimens and quantifications of lymphatic vessel coverage, lymphatic length, and bifurcations of the dorsal and ventral ventricular surfaces upon *Vegfc* conditional elimination with *Wt1^Cre*. (L–O) Representative specimens and quantifications of lymphatic vessel coverage of the dorsal and ventral ventricular surfaces upon *Vegfc* conditional elimination with *Tbx18^Cre*. Statistics: Mann–Whitney test with two-tailed *p*-values. Graphs show mean ± standard deviation. Each dot represents one biological specimen. N = 5 control and 5 mutant hearts in each graph except in (H) and (J), in which 4 control hearts were analyzed, in (N), in which 7 mutant hearts were analyzed an in (O), in which 4 control and 7 mutant hearts were analyzed. Dotted lines in (A), (B), (E), (L), and (M) indicate the limits of the pulmonary artery (in front) and aorta (behind). Dotted lines in (A'), (B'), (E'), (L'), and (M') indicate the limits of the sinus venosus. Scale bars 200 μm.

vessel is always located closer to the epicardium, representing the most superficial coronary vasculature (Fig. 1C',C",E). We thus found that coronary lymphatic vessels do not follow arteries or veins, as it happens in some tissues/organs (Cha et al, 2012; Vaahtomeri et al, 2017), but rather grow freely beneath the epicardium, without following either veins, which grow beneath the lymphatic vasculature, or arteries, which grow within the myocardium. This behavior correlates with the strong expression of VEGFC and CXCL12 in the epicardium (Cavallero et al, 2015; Chen et al, 2014).

While VEGFC is strongly expressed in the epicardium (Chen et al, 2014), it is also expressed in other cardiac cells (Cahill et al, 2021); we therefore specifically studied the relevance of epicardial VEGFC in coronary lymphangiogenesis. We first studied the sensitivity of the developing lymphatic vasculature to increased epicardial VEGFC. For this, we used the *Eef1a1^VegfcGOF* allele (Pichol-Thievend et al, 2018), which provides conditional Cre-mediated VEGFC overexpression, combined with *Wt1^Cre* (Wessels et al, 2012), which drives Cre expression in the epicardium. Hearts with epicardial VEGFC overexpression showed increased coronary lymphatic coverage at E16.5 (Fig. 2A–D), indicating that coronary lymphangiogenesis is sensitive to epicardial VEGFC levels. Next, we studied the requirement for VEGFC function in the epicardium/EPDCs, by crossing *Vegfc^flox/flox* (Lim et al, 2019) with *Wt1^Cre* and analyzing the lymphatic vasculature at E16.5. Homozygous mutant hearts (*Vegfc*-KO) showed 50% reduction in lymphatic coverage in the ventral side of the ventricles and 68% reduction in the dorsal side (Fig. 2E–G). Total lymphatic length and lymphatic vessel bifurcations were also reduced in *Vegfc*-KO hearts (Fig. 2H–K). Epicardial *Vegfc*-KO mice were viable after birth and, as previously described (Liu et al, 2020), we observed a reduction in heart size at postnatal day 20 (P20) (Fig. EV1A).

To confirm these results with another epicardial Cre line, we induced epicardial deletion of *Vegfc* using *Tbx18^Cre* (Cai et al, 2008). Mutant hearts in this case lacked all lymphatic vessels of the dorsal side of the heart and showed only vestigial lymphatic vessels on the ventral side (Fig. 2L–O). These results suggest that the incomplete penetrance observed in the *Wt1^Cre*-deleted hearts is due to lower efficacy of this Cre line compared with *Tbx18^Cre*. Given that at mid-gestation, epicardial recombination mediated by *Tbx18^Cre* and *Wt1^Cre* is similar (Cai et al, 2008; Christoffels et al, 2009; Villa del Campo et al, 2016; Wessels et al, 2012), we studied the level of Cre expression in RNAseq of dissected epicardium/sub-epicardium of E16.5 mutant hearts recombined with either driver. We observed higher Cre expression in *Tbx18Cre* mice than in *Wt1Cre* mice (Fig. EV1B). To determine whether this affected the efficacy in *Vegfc^f* deletion, we performed similar RNAseq analysis in epicardial *Vegfc* mutants. We observed that elimination of *Vegfc* exon 3 was

similarly efficient between *Wt1^Cre*- and *Tbx18^Cre*-recombined hearts at E16.5 (Fig. EV1C). This result suggests that timing of complete elimination of the VEGFC proteins underlies the different affection of the two genetic models. The fact that *Tbx18^Cre* already recombines the pro-epicardial organ (Cai et al, 2008; Tyser et al, 2021), whereas *Wt1^Cre* recombines extensively in the epicardium (Villa del Campo et al, 2016; Wessels et al, 2012) might also contribute to a different timing of VEGFC elimination between the two strains.

Dosed levels of epicardial VEGFC thus appear essential for coronary lymphangiogenesis, with more relevance in the dorsal than in the ventral side of the ventricles. We then studied VEGFR3 expression levels in the E16.5 dorsal and ventral coronary lymphatic vasculature and found no differences (Fig. EV1D,E). To then determine whether the difference between dorsal and ventral lymphangiogenesis could be determined by the signaling environment, we studied VEGFD and Cxcl12 expression by immunofluorescence (Fig. EV1F–I). Whereas we found no differences for VEGFD (Fig. EV1F,G), we observed about 25% higher expression of Cxcl12 on the dorsal side as compared to the ventral side of the heart (Fig. EV1H,I). To more broadly determine whether a different signaling environment might influence dorsal versus ventral coronary lymphangiogenesis, we then compared the transcriptomes of the dorsal and ventral epicardial/subepicardial regions of the E16.5 wild-type heart (Fig. EV1F) and determined the expression levels of 22 known extracellular regulators of lymphangiogenesis (Coso et al, 2014; Hussmann et al, 2023; Trincot et al, 2019; Zheng et al, 2014) (Fig. EV1G). This study showed that mRNAs coding for 8 of these factors were expressed in the ventral epicardium/sub-epicardium at higher levels than in the dorsal, whereas no one of them was expressed at higher levels dorsally (Fig. EV1J,K). This study suggests a generally more lymphangiogenic environment in the ventral subepicardium than in the dorsal one, however, some lymphangiogenic factors, like Cxcl12, appear more abundant dorsally at this stage. *Vegfc* is one of the factors with higher expression in the ventral epicardium, which suggests that the lower availability of this essential lymphangiogenic factor underlies the higher sensitivity of the dorsal lymphatic vasculature to *Vegfc* elimination.

Although less prominent than VEGFC, VEGFD collaborates with VEGFC in promoting lymphangiogenesis in the developing intestinal lymphatic vasculature (Nurmi et al, 2015). Whole-mount detection of VEGFD protein in WT E16.5 hearts revealed expression in the epicardium (Fig. 3A,B) and epicardial-derived cells associated to the growing coronary lymphatic vasculature (Fig. 3C,C'). To functionally assess the role of VEGFD in coronary lymphatic development, we generated *Vegfd* global knockouts using CRISPR-Cas9 deletion of exons 3 and 4, which codify a portion of

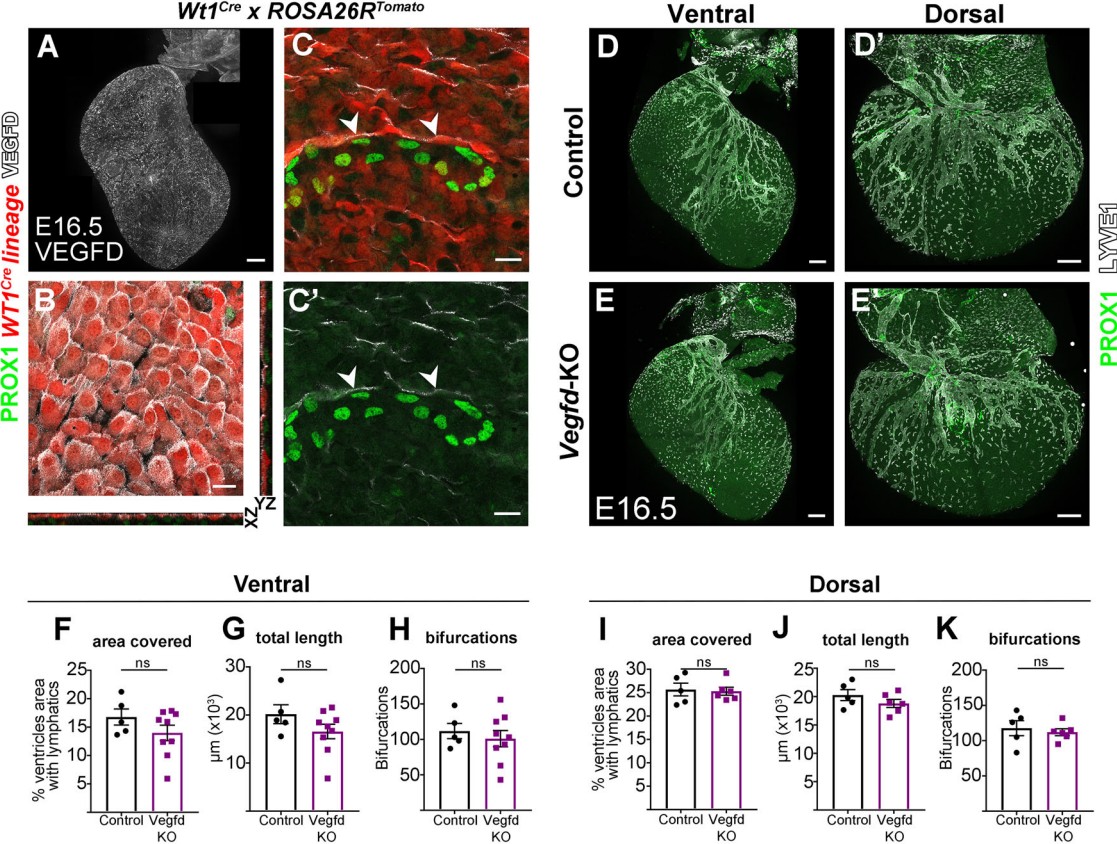

**Figure 3. Epicardial expression of VEGFD is not essential for coronary lymphangiogenesis.**

(A) Confocal detection of VEGFD protein in a whole-mount E16.5 mouse heart (ventral side). This example corresponds to one of the specimens quantified and shown in Fig. EV2A. (B) Maximum projection of three epicardial confocal planes showing detection of VEGFD in the epicardium of a E16.5 *Wt1^Cre^; Rosa26R^Tomato^* mouse heart, in which the *Wt1^Cre^* lineage reveals the epicardium in red. (C–C') Maximum projection of three subepicardial confocal planes showing Prox1⁺ lymphatic endothelial cells surrounded by VEGFD-expressing epicardial-derived cells (arrowheads). (D, E') PROX1 and LYVE1 whole-mount immunostaining of E16.5 control and *Vegfd^−/−^* (*Vegfd*-KO) hearts. (F–K) Quantification of the percentage of area covered by lymphatic vessels on the ventral and dorsal ventricular surfaces (F, I), total lymphatic length (G, J) and number of bifurcations (H, K) in control and *Vegfd*-KO hearts. Statistics: Mann–Whitney test with two-tailed *p*-values. Graphs show mean ± standard deviation. Each point represents a quantified heart. *N* = 5 control and 9 mutant hearts in (F–G) and 5 control and 6 mutant hearts in (I–K). Scale bars: 200 μm in (A, D, E') and 20 μm in (B, C').

VEGFD central receptor-binding domain, "*VEGF homologous domain*" (VHD), essential for its function (Achen et al, 1998) (see Methods). Mice homozygous for this deletion showed a clear reduction of epicardial VEGFD detection by whole-mount immunofluorescence of E16.5 hearts (Fig EV2). As previously described, VEGFD deletion produced viable and fertile animals (Baldwin et al, 2005). The coronary lymphatic vasculature of *Vegfd* mutant hearts at E16.5 showed no significant alterations (Fig. 3D–K), although lymphatic vessel coverage, total lymphatic vessel length and total bifurcations showed a tendency to reduction on the ventral side of the ventricles (Fig. 3F–H).

To determine whether epicardial VEGFD and VEGFC act redundantly in cardiac lymphangiogenesis, we studied the coronary lymphatic vasculature of compound mutants (Fig. 4A–D'). We studied *Wt1^Cre^*-mediated epicardial elimination of *Vefgc* with or without elimination of *Vegfd*. Due to the location of *Vegfd* in the X chromosome, KO males are hemizygous, while KO females are homozygous for the *Vegfd* deletion. Elimination of *Vegfd* in females exacerbated the reduction in coronary lymphatic vasculature observed in the epicardial deletion of *Vegfc* (Fig. 4A–C',E,F). This

reduction was significant only in the ventral side but not on the dorsal side of the ventricles, suggesting a stronger interaction between VEGFC and VEGFD in the ventral side of the heart (Fig. 4E,F). Unexpectedly, we found that *Vegfd* deletion in males did not worsen the reduction in coronary lymphatics observed in the epicardial deletion of *Vegfc* (Fig. 4B,D,D',D',E,F). These results show that epicardial VEGFC and D cooperate in coronary lymphangiogenesis in females but not in males. Previous studies showed a higher lymphatic coverage in females than in males in adult hearts of the C57BL/6 mouse strain (Trincot et al, 2019). To determine whether this is also the case during gestation and in this genetic background, we quantified the lymphatic vessel coverage in control males and females of our VEGFC/VEGFD interaction study. We observed a non-significant tendency to a higher coverage in female hearts compared to male hearts (Fig. EV3A), on both the dorsal and the ventral sides. Sex-differences in coronary lymphatic coverage therefore may start to develop during gestation and are robust to changes in the genetic background. Together with the specific sensitivity of females to VEGFD loss, these observations suggest that enhanced coronary lymphangiogenesis might be driven

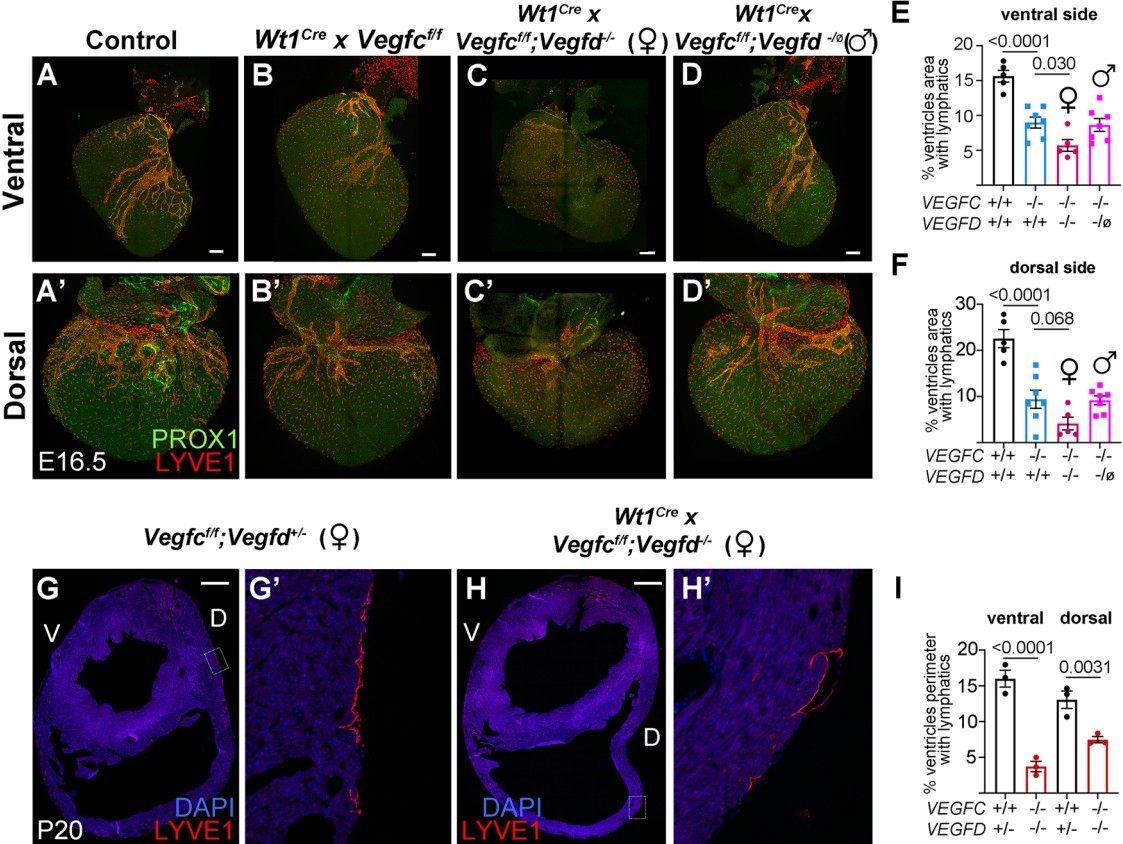

**Figure 4. Sex-specific cooperative redundant roles of VEGFC and VEGFD in coronary lymphangiogenesis.**

(A–D') Representative E16.5 specimens showing lymphatic vessel coverage of the dorsal and ventral ventricular surfaces upon combined conditional elimination of *Vegfc* with *Wt1Cre* and constitutive elimination of *Vegfd*. (E, F) Quantification of the ventricular lymphatic vessel coverage in combinations of *Vegfc/d* mutants. Statistics: one-way ANOVA with Sidak's correction for multiple comparisons. Graphs show mean ± standard deviation. Each dot represents one biological specimen. Scale bars: 100 μm. (G–H') Representative sections of P20 specimens showing lymphatic vessel coverage of the dorsal and ventral ventricular surfaces upon combined conditional elimination of *Vegfc* with *Wt1Cre* and constitutive elimination of *Vegfd*. (G') and (H') represent the boxed regions in (G) and (H), respectively. (I) Quantification of the ventricular lymphatic vessel coverage in combinations of *Vegfc/d* mutants at P20. Statistics: one-way ANOVA with Sidak's correction for multiple comparisons. Graphs show mean ± standard deviation. Each dot represents one biological specimen. N = 5 wild type, 7 Vegfc−/−, 5 female *Vegfc/d* double KO and 7 male *Vegfc/d* double KO hearts in (E) and (F). N = 3 *Vegfd*+/− and 3 *Vegfc/d* double KO hearts in (I). Scale bars: 1 mm. D: dorsal; V: ventral. Significant P-values are shown on the graphs. When not shown, comparsions were not significant (P-value > 0.05).

by VEGFD signaling in females. To study the possible bases of the differential coronary lymphangiogenesis in males and females, we then studied the transcriptome of the epicardial/subepicardial region in E16.5 wild type female and male hearts. Whereas a large number of transcripts was found differentially expressed between males and females—including the expected male-specific Y-linked genes and female-specific X-inactivation genes—we did not observe differential expression for any of 22 known lymphangiogenic factors, including VEGFC and D (Fig. EV3B). These results suggest that the differences in coronary lymphatic vasculature between females and males are not dependent on differences in the sub-epicardial lymphangiogenic environment.

Finally, to determine whether the coronary lymphatic vasculature develops postnatally in the VEGFC/D mutant models generated, we studied epicardial VEGFC/D KO female hearts at P20. We found that the ventral coronary lymphatic vasculature suffers a 77% reduction at this stage, suggesting continuous requirement of epicardial VEGFC/D, whereas the dorsal

vasculature shows a 41% reduction, suggesting partially postnatal recovery (Fig. 4G–I).

Our results thus reveal an essential role of the epicardium in coronary lymphangiogenesis during gestation and postnatally through the production of VEGFC and VEGFD. Previously, elimination of epicardial VEGFC showed delayed dorsal blood coronary vasculature development (Chen et al, 2014), however, here we show complete agenesis of the lymphatic vasculature, which reveals regulation of lymphangiogenesis by epicardial VEGFC and VEGFD independently of the blood vasculature. While we used epicardial drivers, epicardial-derived cells inherit this recombination, they associate intimately with LECs, and we showed that they express VEGFD. We cannot, therefore, exclude a role of epicardial-derived cells in contributing to coronary lymphangiogenesis. Nonetheless, the epicardium/subepicardium appears to be the most important driver of lymphangiogenesis, given that epicardial-derived cells invade both the sub-epicardium and the myocardium, whereas lymphatic vessels always grow in

intimate contact with the epicardium, without entering the myocardium. This view is in agreement with the strong expression of the lymphatic endothelial cell guidance cytokine CXCL12 in the epicardium, together with VEGFC and VEGFD (Cavallero et al, 2015; Chen et al, 2014). Interestingly, the CXCL12/CXCR4 axis has recently been shown important for lymphangiogenesis and able to regulate the levels of VEGFR3 in LECS (Do et al, 2024). Another potential epicardial-derived cell population contributing to lymphangiogenesis through VEGFC/D production is the smooth muscle of the coronary vessels, however, single-cell transcriptomics has shown that smooth muscle cells express low levels of VEGFC compared to epicardium and fibroblasts (Feng et al, 2022). Furthermore, smooth muscle of the coronary vessels at the base of the ventricles mostly originates from neural crest and second heart field and therefore, their influence on lymphangiogenesis through VEGFC/D production would be intact in epicardial mutants; nonetheless, coronary lymphatics are fully eliminated from the base of the ventricles in $Tbx18^{Cre}$ VEGFC mutants. These considerations, together with the lack of association of lymphatic vessels with great coronary blood vessels and their subepicardial location during pre-natal heart development strongly suggest that vascular smooth muscle does not play a significant role in the establishment of coronary lymphangiogenesis.

We also uncovered a sex-specific requirement for VEGFD in the heart, however, the role of VEGFD is only revealed in a defective VEGFC background, indicating that VEGFC alone is sufficient to drive coronary lymphangiogenesis in both males and females. Why males are insensitive to VEGFD deletion is difficult to characterize and may relate to sex-specific sensitivity to VEGFD signaling.

Finally, we report that the ventral subepicardium appears to be a more lymphangiogenic environment than the dorsal one, which may underlie differential affection of dorsal and ventral lymphatic vessels to mutations. The apparently different requirements of VEGFC and VEGFD in ventral versus dorsal lymphangiogenesis may also relate to the recent discovery of the contribution of second heart field precursors exclusively to the ventral coronary lymphatic endothelium (Lioux et al, 2020). The different origins of LECs in the ventral versus the dorsal side of the ventricles may determine intrinsic differential sensitivities of the ventral and dorsal lymphatic vasculatures to VEGFC and VEGFD, however, here we could not relate this differential sensitivity to differential expression of VEGFR3, so further studies will be required to elucidate this aspect.

In summary, we report a cardiac-specific mechanism of lymphatic vasculature development that relies on essential and redundant contributions of epicardial VEGFC and VEGFD to coronary lymphangiogenesis.

# Methods

### Reagents and tools table

| Reagent/Resource | Reference or Source | Identifier or Catalog Number |
| --- | --- | --- |
| **Experimental models** | | |
| $Wt1^{Cre}$ (*Mus musculus*) | Wessels et al, 2012 | N/A |
| $Tbx18^{Cre}$ (*Mus musculus*) | Cai et al, 2008 | MGI:3801041 |

| Reagent/Resource | Reference or Source | Identifier or Catalog Number |
| --- | --- | --- |
| $Rosa26^{tdtmt}$ (*Mus musculus*) | Madisen et al, 2010 | MGI:3809523 |
| $Vegfc^{flox}$ (*Mus musculus*) | Lim et al, 2019 | N/A |
| $Eef1a1^{VegfcGOF}$ (*Mus musculus*) | Pichol-Thievend et al, 2018 | N/A |
| $Vegfd^{-}$ (*Mus musculus*) | This study | Methods and Protocols |
| **Antibodies** | | |
| CX-40 | Alpha Diagnostics | CX40-A |
| EMCN eFluor660 | eBioscience™ | 50-5851-82 |
| LYVE1 | ReliaTech | 03-PA50S |
| PROX1 | R&D systems | AF2727 |
| SM22α | Abcam | ab14106 |
| VEGFD | Cusabio CSB- | CSB-PA07554A0Rb |
| VEGFR3 | R&D Systems | AF743 |
| CXCL12 | R&D Systems | MAB350 |
| Goat anti-Rabbit 405 | Thermo Fisher | A31556 |
| Goat anti-Rabbit 488 | Life Technologies™ | A11034 |
| Donkey anti-Rabbit 488 | Molecular Probes | A21206 |
| Goat anti-Rabbit 633 | Life Technologies™ | A21071 |
| Donkey anti-Rabbit 633 | Sigma | SAB4600132 |
| Donkey anti-Rabbit Cy5 | Jackson Immuno | 711-495-152 |
| Goat anti-Rat 488 | Thermo Fisher | A11006 |
| Donkey anti-Rat 488 | Molecular Probes | A21208 |
| Donkey anti-Rat Cy5 | Jackson Immuno | 712-175-150 |
| Donkey anti-Goat 488 | Thermo Fisher | A11055 |
| Donkey anti-Goat 647 | Thermo Fisher | A21447 |
| Goat anti-Mouse 633 | Life Technologies™ | A21052 |
| **Oligonucleotides and other sequence-based reagents** | | |
| Fw flanking Vegfd exon 3 | This study | Methods and Protocols |
| Rev flanking Vegfd exon 3 | This study | Methods and Protocols |
| Fw flanking Vegfd exon 4 | This study | Methods and Protocols |
| Rev flanking Vegfd exon 4 | This study | Methods and Protocols |
| Fw flanking Vegfd exons 3 and 4 | This study | Methods and Protocols |

| Reagent/Resource | Reference or Source | Identifier or Catalog Number |
|---|---|---|
| Rev flanking Vegfd exons 3 and 4 | This study | Methods and Protocols |
| sgRNA targeted to Vegfd exon 2 | This study | Methods and Protocols |
| sgRNA targeted to Vegfd exon 2 | This study | Methods and Protocols |
| sgRNA targeted to Vegfd exon 4 | This study | Methods and Protocols |
| sgRNA targeted to Vegfd exon 4 | This study | Methods and Protocols |
| **Chemicals, Enzymes and other reagents** | | |
| Evolve-KSOM medium | Zenith Biotech | ZEKS-050 |
| S.p. HiFi Cas9 Nuclease V3 | IDT Alt-R® | 1081060 |
| KCl | Sigma | P9541 |
| Sucrose | Sigma | 16104 |
| Gelatin | Sigma | G2500 |
| Isopentane | Sigma | 1060561000 |
| Triton X-100 | Sigma | T9284 |
| TNB blocking reagent | Perkin-Elmer | FP1012 |
| Tween-20 | Sigma | P9416 |
| Fluorescence mounting medium | Dako | s3023 |
| RNeasy Micro kit | Qiagen | 74004 |
| NEBNext Ultra RNA Library preparation kit | New England Biolabs | NEB #E7770 |
| **Software** | | |
| ImageJ | https://imagej.net/ij/ | Version 1.54 g |
| bcl2fastq | Illumina | v2.20.0.422 |
| FastQC | http://www.bioinformatics.babraham.ac.uk/projects/fastqc/ | Version 0.12.0 |
| Cutadapt | Martin, 2011 | Version 1.7.1 |
| RSEM | https://github.com/deweylab/RSEM | v1.2.20 |
| Bioconductor package LIMMA | Ritchie et al, 2015 | Version 3.20 |

## Mouse lines

Animals were handled in accordance with CNIC Ethics Committee, Spanish laws and the EU Directive 2010/63/EU for the use of animals in research. All mouse experiments were approved by the CNIC and Universidad Autónoma de Madrid Committees for "Ética y Bienestar Animal" and the area of "Protección Animal" of the Community of Madrid with references PROEX 220/15 and PROEX 144.1/21 For this study, mice were maintained on a mixed background. Mouse experiments performed at Northwestern University were performed in accordance with protocols approved by Northwestern University Institutional Animal Care and Use Committee. The mouse alleles used here and already described were: $Wt1^{Cre}$ (Wessels et al, 2012), $Tbx18^{Cre}$ (Cai et al, 2008), $Rosa26^{tdtmt}$ (Madisen et al, 2010), $Vegfc^{flox}$ (Lim et al, 2019), $Eef1a1^{VegfcGOF}$ (Pichol-Thievend et al, 2018).

A new *Vegfd knockout* line was generated using CRISPR-Cas9 technology. Four sgRNAs that recognized *Vegfd* genomic sequences were designed using the CRISPOR web tool (Concordet and Haeussler, 2018): two sgRNAS to target intron 2 (A and B), and two to target intron 4 (C and D). sgRNA-coding sequences (A) TAGGT-TAAGTTCCCATATAGTGG; (B) GCGTCATGAAAAGCATGT-CAGGG; (C) ATGCCTGTATAATGGGTAAAGG; (D) GTGCAACA CATGTCTTTCTG AGG. The four sgRNAs were used simultaneously. Around 3.4 kb of *Vegfd* gene from intron 2 to intron 4 and including exon 3, intron 3 and exon 4 were deleted. Exons 3–4 code for VEGFD amino acids 106 to 218. To generate mouse mutants carrying the deletion ten 3 to 5-weeks old C57BL/6JCrl females were superovulated by injection of 5 IU of PMSG and 48 h later with 5 IU of hCG. Females were then crossed with C57BL/6JCrl males and the next morning they were checked for positive plug. Fertilized zygotes were retrieved and incubated at 37 °C with 5% $CO_2$/5% $O_2$ in Evolve-KSOM medium (Zenith Biotech ZEKS-050) for pronuclear microinjection of 1–2 pL containing 100 µg of Cas9 protein (IDT Alt-R® S.p. HiFi Cas9 Nuclease V3, 1081060) and 0.305 µM of each sgRNA (A to D) (IDT Alt-R® CRISPR-Cas9 sgRNA). Injected embryos were then incubated in Evolve-KSOM medium at 37 °C and 5% $CO_2$/5% $O_2$ overnight. The next morning, the embryos at two-cell stage were transferred to CD-1 pseudopregnant females. 5 microinjected animals were obtained at weaning and 3 of them had exons 3 and 4 deleted and were used to establish the mouse line. For genotyping, the following primers were used: To detect the deletion of Exon 3: Fw1: GTGCTATCCAGCTGTAGCCT; Rv1: CCCCTGAGCCTGTTTCTT-TACT; To detect the deletion of Exon 4: Fw2: GGGCAAAAATGCA-GATGGTGG; Rv2: GATCCTCAAGGTTTTGGGTCCT; To detect the total deletion of both, exon 3 and exon 4: Fw1: GTGCTATC-CAGCTGTAGCCT; Rv2: GATCCTCAAGGTTTTGGGTCCT. The specific deletions obtained were characterized by Sanger DNA sequencing. We confirmed that the mutant sequences aligned with the flanking regions of *Vegfd* exons 3 and 4 and contained deletions from 276 bp upstream exon 3 to 67 bp downstream exon 4 (2 founders) or from 272 bp upstream exon 3 to 90 bp downstream exon 4 (1 founder). Females were maintained in heterozygosity and the males in hemizygosity. These animals were crossed and maintained in a C57BL/6 background. Mouse strains described here are available upon request.

Except for *Vegfd*-KO line, mice were genotyped by PCR as described in the original reports. Male and female mice older than 8 weeks of age were used for mating. Experimental specimens were retrieved during gestation and sex-determined by PCR for *Vegfd*-KO and *Vegfd*-KO;*Vegfc*-KO embryos.

## Embryo and organ retrieval

The morning in which the vaginal plug was detected was considered as embryonic day 0.5 (E0.5). Pregnant females were sacrificed by $CO_2$ inhalation followed by cervical dislocation. Embryos were dissected in PBS with a sprinkle of Heparin (ROVI 1000 IU/mL). Fetuses were decapitated and placed in a 50 mM KCl (Sigma P9541) solution in PBS with heparin to stop the hearts in diastole and avoid clotting before dissection. Hearts were dissected

and fixed in PFA 4% in PBS overnight at 2 °C. The tip of the tail was used to genotype specimens.

## Tissue sectioning and immunostaining

Hearts were washed in PBS after fixation and cryoprotected in 15% sucrose (Sigma, 16104) PBS overnight at 4 °C. The next morning, sucrose was removed and a 37 °C pre-heated solution of 7.5% gelatin (Sigma, G2500), 15% sucrose in PBS was added. Hearts were incubated in this solution for at least 4 h and then allowed to solidify at 4 °C overnight. Gelatin blocks were snap-frozen in a −70 °C solution of isopentane (Sigma, 1060561000) for 1 min. The frozen blocks were kept at −80 °C until sectioned. 8-μm-thick cryo-sections were made using a Leica CM1950 automated Cryostat and stored at −20 °C until used. Slides were thawed at room temperature and gelatin was removed from the slides by two 10-min washes with PBS at 37 °C and a quick wash with room-temperature (RT) PBS. Sections were then permeabilised with 0.5% Triton X-100 (Sigma T9284) in PBS for 30 min at RT, followed by PBS washing and treated with TNB blocking reagent (Perkin Elmer, FP1012) for 1 h at RT. The sections were then incubated with primary-antibody dilutions prepared in the same TNB blocking solution within a humid chamber at 4 °C overnight. Secondary antibodies were incubated for 1 h at RT. Slides were washed with 0.01% Tween-20 (Sigma P9416) in PBS several times between the previous steps. Slides were mounted with Dako fluorescence mounting medium (s3023).

## Antibodies

CX-40: Alpha Diagnostics CX40-A, Rabbit 1:200; EMCN eFluor660 eBioscience™ 50-5851-82, Rat 1:100/1:200; LYVE1 ReliaTech 03-PA50S, Rabbit 1:200; PROX1 R&D systems AF2727, Goat 1:200; SM22α abcam ab14106, Rabbit 1:200; VEGFD Cusabio CSB-PA07554A0Rb, Rabbit 1:200; VEGFR3 R&D Systems AF743, Goat 1:100/1:200; CXCL12 R&D Systems MAB350, Mouse 1:100.

## Whole-mount immunofluorescence

For hearts, after fixation and subsequent wash with PBS, all the following steps were performed at 4 °C in a rotating wheel. 500 μL were added at each step, except for the washes that were performed in a 50 ml Falcon tube. Protocol: Permeabilization for 2 days in 0.5% Triton X-100 in PBS was followed by 2-h wash with PBS. TNB blocking was performed over day and primary antibody incubation for 3 days followed by several washes over day with 0.01% Tween 20 in PBS. Secondary antibodies were incubated from overnight to 2 days, depending on the antigen and cardiac stage. Finally, hearts were thoroughly washed with 0.01% Tween 20 in PBS followed by PBS to remove detergent. To improve image acquisition while preserving fluorescence, hearts were mildly clarified with increasing glycerol concentrations starting at 20% glycerol in PBS up to 80%. For immunofluorescences of Fig. 2, the blocking solution was 3% BSA, 0.1% Triton, and 5% donkey serum in PBS.

## Image acquisition and quantification

Whole-mount and cardiac sections immunofluorescence were acquired with a Nikon W1 Spinning Disk inverted confocal microscope for Fig. 2A–E'. For the rest of the images, a Leica TCS

SP8 coupled to a DMi8 inverted microscope was used with Navigator module and equipped with a white light laser. Images were analyzed and quantifications were made using ImageJ (http://rsb.info.nih.gov/ij). Maximum projections were acquired using TileScan and z-stack functions. The Maximum Projection of equivalent z-stacks for the different hearts was used for quantification. We used the Freehand selection tool of ImageJ to select the whole ventral or dorsal ventricular surfaces and the area covered by Lymphatic vessels. For LYVE1 whole-mount quantifications of Vegfd-KOs, Vegfc-GOF and VEGFC-KO;VEGFD-KO hearts, a macro was used to quantify lymphatic coverage. This included several steps: duplicate channel of interest, median filter, binary conversion to Mask, analyze particles tool to remove smaller particles (macrophages) and finally, after creating a raw selection, manual curation using the Freehand selection tool. To quantify the total lymphatic length the Freehand line tool was used to draw lines running medial along each lymphatic branch. Total length was obtained by adding the length of every branch. Bifurcations were manually counted, considering each intersection point of two or more lymphatic branches.

## RNAseq analysis

The epicardium/subepicardium was dissected as previously described (Lioux et al, 2020) and quickly frozen in separate 1.5 ml Eppendorf tubes on dry ice. For mRNA isolation, samples from at least three different litters were used. As low RNA amount from one single epicardial layer was expected, two dissected epicardia of the same genotype were pooled together. Therefore, the three biological replicates per genotype represent six different hearts pooled by pairs. RNA was then purified from each pool using RNeasy Micro kit (Qiagen 74004). 40 ng of total RNA were used to generate barcoded RNA-seq libraries using the NEBNext Ultra RNA Library preparation kit (New England Biolabs). Briefly, poly A + RNA was purified using poly-T oligo-attached magnetic beads followed by fragmentation and then first and second cDNA strand synthesis. Next, cDNA 3' ends were adenylated and the adapters were ligated followed by PCR library amplification. Finally, the size of the libraries was checked using the Agilent 2100 Bioanalyzer DNA 1000 chip and their concentration was determined using the Qubit® fluorometer (Life Technologies). Libraries were sequenced on a HiSeq2500 (Illumina) to generate 60 bases single reads and processed with RTA v1.18.66.3. FastQ files for each sample were obtained using bcl2fastq v2.20.0.422 software (Illumina). The raw data analysis of RNA-seq data were generated by CNIC Bioinformatics unit. Read quality was assessed with FastQC (http://www.bioinformatics.babraham.ac.uk/projects/fastqc/). Illumina adapter sequences were trimmed with Cutadapt 1.7.1 (Martin, 2011), which also discarded reads that were shorter than 30 bp. The resulting reads were mapped against the mouse transcriptome (GRCm38, release 91; aug2017 archive) and quantified using RSEM v1.2.20. Data were then processed with a differential expression analysis pipeline that used Bioconductor package LIMMA (Ritchie et al, 2015) for normalization and differential expression testing. Genes with at least 1 count per million in at least 3 samples were considered for statistical analysis. We considered as differentially expressed those genes with Benjamin-Hochberg adjusted $p$-value < 0.05. Fold change and log (ratio) values were

calculated to represent gene expression differences between conditions. For pathways analysis, the Ingenuity Pathway Analysis software was used.

## Statistics

The details of each test used are specified in Figure legends. As a general rule, Mann–Whitney two-tailed test was performed to compare two groups of quantitative data. One-way ANOVA was performed to compare more than two groups of quantitative data with one independent variable, after a normality test (Kolmogorov–Smirnov) was found positive for all samples. Data is indicated as mean ± SEM of the individual plotted values. All comparisons and graphs were made using GraphPad Prism 9 statistical analysis software. In all cases *P*-values were two-tailed and adjusted by Sidak's correction for multiple measurements. Values of $p \geq 0.05$ were considered non-significant. No blinding was used in the experiments.

## Data availability

Sequencing data have been deposited to GEO with accession number GSE284800. Source data are available at Mendeley (https://data.mendeley.com/datasets/4dhff9k2xb/2).

The source data of this paper are collected in the following database record: biostudies:S-SCDT-10_1038-S44319-025-00431-7.

## Peer review information

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

## Acknowledgements

We thank Cristina Villa and the Torres group for helpful comments and discussions, Xiaolei Liu and Michael Oxendine for advice during experimentation, Wanshu Ma for the *Eef1a1^VegfcGOF* mouse strain, Wanshu Ma and Mark Kahn for the *Vegfc^flox/flox* mouse, and the CNIC Genomics, Microscopy, Cellomics and Transgenesis Units personnel for their support to this work. This work was supported by the European Commission H2020 Program grant SC1-BHC-07-2019. Ref. 874764 "REANIMA" to MT; the Spanish Ministerio de Ciencia e Innovación grant PGC2018-096486-B-I00 to MT; Grant TNE-17CVD04 from the Leducq Foundation to MT; Comunidad de Madrid grant P2022/BMD-7245 to MT; RO1HL151388 and RO1HL162800 to GO, FPU grant from the Spanish Ministry of Education, Culture and Sports (Grant FPU15/02955) and EMBO Short-Term Fellowship number 8357 to EdC; for experiments in the Unidad de Microscopía e Imagen Dinámica, CNIC, ICTS-ReDib, MCIN/AEI /10.13039/501100011033 and FEDER "Una manera de hacer Europa" (#ICTS-2018-04-CNIC-16). The CNIC is supported by the Ministerio de Ciencia e Innovación and the Pro CNIC Foundation, and is a Severo Ochoa Center of Excellence (Grant number CEX2020-001041-S funded by MICIU/AEI/10.13039/501100011033).

## Author contributions

**Ester de la Cruz**: Conceptualization; Formal analysis; Investigation; Methodology; Writing—original draft; Writing—review and editing. **Vanessa Cadenas**: Methodology. **Susana Temiño**: Methodology. **Guillermo Oliver**: Conceptualization; Resources; Supervision; Writing—review and editing. **Miguel Torres**: Conceptualization; Resources; Formal analysis; Supervision; Funding acquisition; Investigation; Writing—original draft; Writing—review and editing.

Source data underlying figure panels in this paper may have individual authorship assigned. Where available, figure panel/source data authorship is listed in the following database record: biostudies:S-SCDT-10_1038-S44319-025-00431-7.

## Disclosure and competing interests statement

The authors declare no competing interests.

# Expanded View Figures

**Figure EV1.  Characterization of epicardial roles in Lymphangiogenesis.**

(**A**) Heart weight to body weight ratios (H/BW) in Control (*Vegf*^f/f^) and epicardial *Vegfc* KO (*WT1*^Cre^;*Vegf*^f/f^) hearts at postnatal day 20 (P20). (**B**) RNAseq quantification of Cre mRNA expression in dissected epicardial/subepicardial region from *Tbx18*^Cre^or *WT1*^Cre^ hearts. CPMs: counts per million. (**C**) RNAseq quantification of *Vegfc* mRNA exon 3 expression in dissected epicardial/subepicardial region from Control, *Tbx18*^Cre^; *Vegfc*^f/f^ or *WT1*^Cre^; *Vegfc*^f/f^ hearts (RPKM: Reads Per Kilobase Million). Statistics in (**A**), (**B**), according to the linear modeling in the LIMMA analysis software for RNAseq analysis (see Methods). Benjamini-Hochberg adjusted *P*-values are shown. Graphs show mean ± standard deviation. Each dot represents one biological specimen. $N = 6$ control and 5 *Vegfc*−/− hearts in (**A**) and (**B**). Statistics in (**C**): one-way ANOVA with Sidak's correction for multiple comparisons. Graphs show mean ± standard deviation. Each dot represents one biological specimen. $N = 8$ control, 3 *WT1*^Cre^;*Vegf*^f/f^ and 5 *Tbx18*^Cre^; *Vegfc*^f/f^ hearts. (**D**) Detection of VEGFR3 expression by whole-mount immunofluorescence in wild type E16.5 Hearts. (**E**) Quantification of the mean intensity of VEGFR3 signal in coronary lymphatic vessels of the dorsal and ventral sides of the hearts shown in (**D**). (**F**) Detection of VEGFD expression by whole-mount immunofluorescence in wild type E16.5 Hearts. (**G**) Quantification of the mean intensity of VEGFD signal in coronary lymphatic vessels of the dorsal and ventral sides of the hearts shown in (**F**). (**H**) Detection of Cxcl12 expression by whole-mount immunofluorescence in wild type E16.5 Hearts. (**I**) Quantification of the mean intensity of Cxcl12 signal in coronary lymphatic vessels of the dorsal and ventral sides of the hearts shown in (**H**). Statistics in (**E**, **G**, **I**): Wilcoxon matched-pairs signed rank test with two-tailed *P*-values. Non-significant results are not indicated. Graphs show mean ± standard deviation. Each dot represents one biological specimen. $N = 4$ ventral sides and 5 dorsal sides in (**E**); 5 ventral and dorsal sides in (**G**) and 12 ventral and dorsal sides in (**I**). (**J**) Volcano plot representing the results of RNAseq comparing dorsal and ventral dissected epicardial/subepicardial regions from wild type E16.5 hearts. Genes encoding lymphangiogenic factors and differentially expressed appear highlighted in red. (**K**) Specific results from the RNAseq analysis shown in (**J**) for 22 selected genes encoding lymphangiogenic factors. Statistics in (**J**, **K**), according to the linear modeling in the LIMMA analysis software for RNAseq analysis (see Methods). Benjamini- Hochberg adjusted *P*-values are shown. When not shown, comparsions were not significant (*P*-value > 0.05). Scale bars 100 μm. Each dot represents a biological replicate. $N = 3$ ventral and 3 dorsal sides for all gene expressions represented.

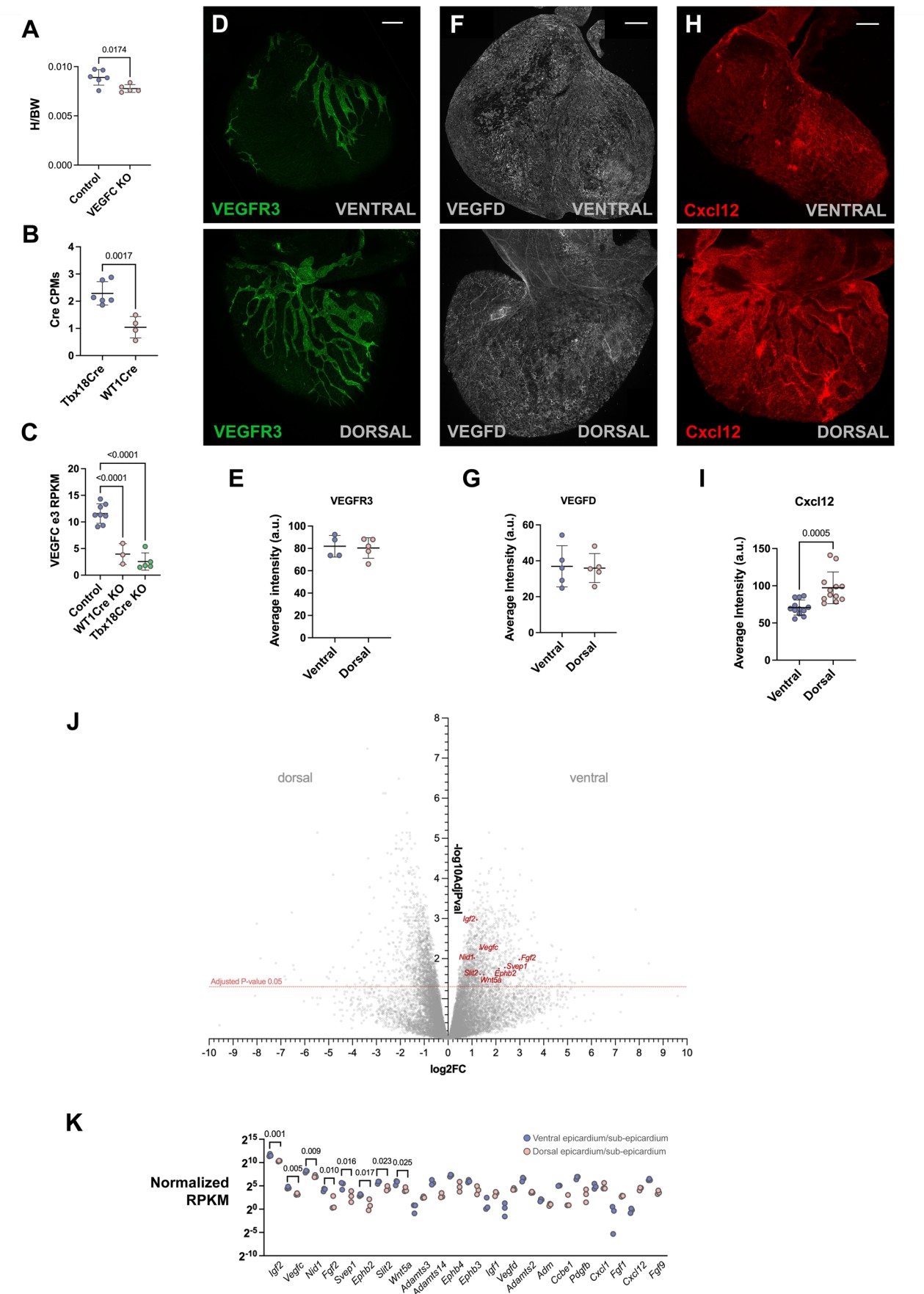

Ventral   Dorsal

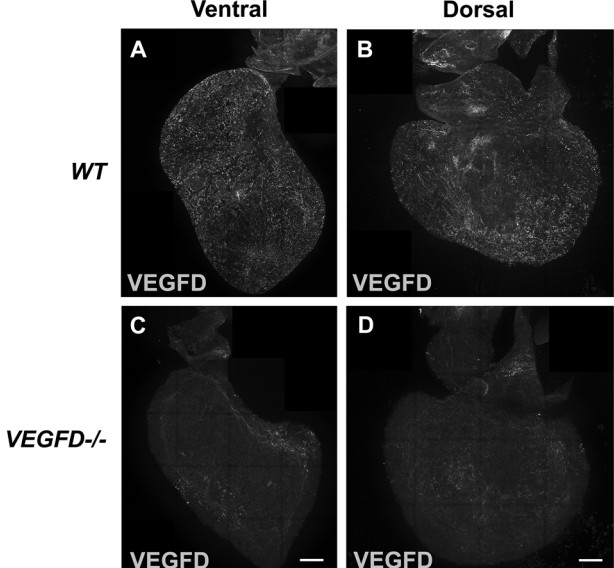

*WT*

*VEGFD-/-*

**Figure EV2.   Characterization of VEGFD epicardial expression.**

(**A**, **B**) Detection of VEGFD expression by whole-mount immunofluorescence in wild type E16.5 hearts in ventral (**A**) and dorsal (**B**) views. The specimen in (**A**) is also shown as an example of the VEGFD expression pattern in Fig. 3A. (**C**, **D**) Equivalent detections in *Vegfd* knockout hearts. $N = 5$ Control and 2 VEGFD KO hearts. Scale bars 100 μm.

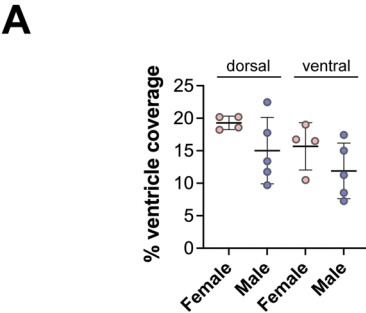

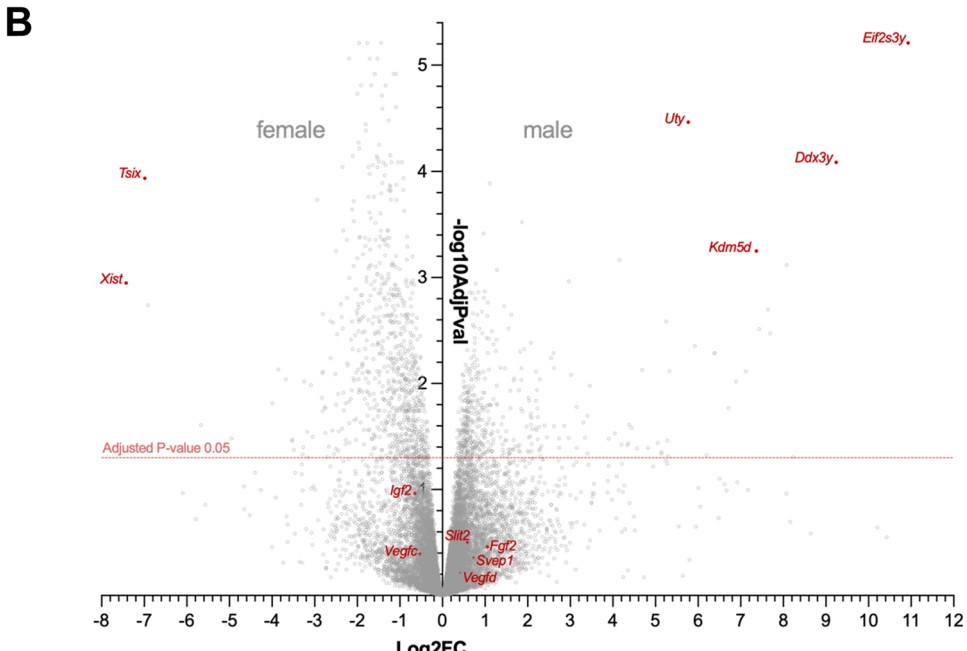

**Figure EV3.  Analysis of sex-specific features of coronary lymphangiogenesis.**

(**A**) Quantification of lymphatic coverage on the ventricles of wild type E16.5 female and male hearts. Statistics: One-way ANOVA with Sidak's multiple comparison test (non-significant). Graphs show mean ± standard deviation. Each dot represents one biological specimen. $N = 4$ female and 5 male samples for both the dorsal and the ventral sides. (**B**) Volcano plot representing the results of RNAseq comparing female to male dissected epicardial/subepicardial regions from wild type E16.5 hearts. Genes encoding for Y-linked and X-chromosome inactivation-specific genes appear in red. Some lymphangiogenic factors showing non-significant differential expression are highlighted in red. Statistics in (**B**), according to the linear modeling in the LIMMA analysis software for RNAseq analysis (see Methods). Benjamini-Hochberg adjusted *P*-values are shown.

