## [Peer Review File · EMBO Reports]

Epicardial VEGFC/D signaling is essential for coronary lymphangiogenesis

Ester de la Cruz, Vanessa Cadenas, Susana Temiño, Guillermo Oliver, and Miguel Torres

Corresponding author(s): Miguel Torres (mtorres@cnic.es)

Review Timeline:

Submission Date:	10th Feb 23
Editorial Decision:	4th Apr 23
Revision Received:	20th Dec 24
Editorial Decision:	4th Feb 25
Revision Received:	8th Feb 25
Accepted:	3rd Mar 25

Editor: Ioannis Papaioannou / Deniz Senyilmaz Tiebe

Transaction Report:

Dear Dr. Torres,

Thank you for submitting your research manuscript for consideration by EMBO reports and for your patience during peer review. Your manuscript has now been seen by three experts in the field, and we have received the full set of their reports, which are included below.

As you will see, the referees acknowledge that the findings are novel and potentially interesting, and they commend the generation of mouse models for characterizing the regulation of cardiac lymphangiogenesis during development. However, they also identify limitations in the study and raise a number of concerns. Referee #1 points out that better characterization of the differences observed between *Wt1-cre-* and *Tbx18-cre-*driven *Vegfc* deletion, as well as of the VEGF-D knockout line would substantially strengthen the manuscript. Referee #2 suggests that some relevant references should be added and that the study should be better placed into context, while they also further recommend the analysis of the protein levels of lymphangiogenic growth factors in male vs. female wild-type mouse dorsal vs. ventral epicardium, as well as the investigation of the possibility that epicardial-deleted *Vegfc* embryos develop cardiac lymphatics postnatally. Referee #3 further suggests deletion of *Vegfc* using *Tie2-Cre* and *SMA-Cre*. In addition, the referees provide a number of additional suggestions for the improvement of the study and the manuscript.

Given these constructive comments, we would like to invite you to revise your manuscript with the understanding that the referee concerns (as detailed above and in their reports) must be fully addressed and their suggestions taken on board. Please address all referee concerns in a complete point-by-point response. Acceptance of the manuscript will depend on a positive outcome of a second round of review. It is EMBO reports policy to allow a single round of revision only and acceptance or rejection of the manuscript will therefore depend on the completeness of your responses included in the next, final version of the manuscript. If you have any questions or comments, we can also discuss the revisions in a video chat, if you like.

We realize that it is difficult to revise to a specific deadline. In the interest of protecting the conceptual advance provided by the work, we usually recommend a revision within 3 months (July 3rd). Please discuss with me the revision progress ahead of this time if you require more time to complete the revisions.

Please note that you can publish your study either as a Report or as an Article. For Reports, the manuscript should not exceed 27,000 characters (including spaces but excluding Materials & Methods and References) and 5 main plus 5 Expanded View figures. The Results and Discussion sections must be combined, which will help to shorten the manuscript text by eliminating some redundancy that is inevitable when discussing the same experiments twice. For an Article there are no length limitations, but it should have more than 5 main figures and the Results and Discussion sections must be separate. In both cases, the entire Materials and Methods must be included in the main manuscript file.

IMPORTANT NOTE:

We perform an initial quality control of all revised manuscripts before re-review. Your manuscript will FAIL this control and the handling will be DELAYED if the following APPLIES:

- 1) If a data availability section providing access to data deposited in public databases is missing. If you have not deposited any data, please add a sentence to the data availability section that explains that (see below for more information).
- 2) If your manuscript contains statistics and error bars based on $n=2$. Please use scatter plots in these cases. No statistics should be calculated if $n=2$.

- 1) A .docx formatted version of the manuscript text (including legends for main figures, EV figures and tables). Please make sure that the changes are highlighted to be clearly visible.
- 2) Individual production quality figure files as .eps, .tif, .jpg (one file per figure). Please download our Figure Preparation Guidelines (figure preparation pdf) from our Author Guidelines pages <https://www.embopress.org/page/journal/14693178/authorguide> for more info on how to prepare your figures.
- 3) A .docx formatted letter INCLUDING the reviewers' reports and your detailed point-by-point responses to their comments. As

part of the EMBO Press transparent editorial process, the point-by-point response is part of the Review Process File (RPF), which will be published alongside your paper unless you opt out of this (please see below for further information).

4) A complete author checklist, which you can download from our author guidelines (). Please insert information in the checklist that is also reflected in the manuscript. The completed author checklist will also be part of the RPF (please see below for more information).

5) Please note that all corresponding authors are required to supply an ORCID ID for their name upon submission of a revised manuscript (). Please find instructions on how to link your ORCID ID to your account in our manuscript tracking system in our Author guidelines
()

6) We replaced Supplementary Information with Expanded View (EV) Figures and Tables that are collapsible/expandable online. A maximum of 5 EV Figures can be typeset. EV Figures should be cited as 'Figure EV1, Figure EV2' etc... in the text and their respective legends should be included in the main text after the legends of regular figures.

7) Please note that a "Data availability" section at the end of Materials and Methods is now mandatory. In case you have no data that require deposition in a public database, please state so instead of refereeing to the database: "Our study includes no data deposited in public repositories." under the heading "Data availability".
See also). Please note that the Data availability statement is restricted to new primary data that are part of this study.

8) We request authors to consider both actual and perceived competing interests. Please review the new policy () and update your competing interests statement if necessary. Please name this section 'Disclosure and competing interests statement' and place it after the Acknowledgements section.

9) Figure legends and data quantification:
The following points must be specified in each figure legend:

- the name of the statistical test used to generate error bars and P values,
- the number (n) of independent experiments (please specify technical or biological replicates) underlying each data point,
- the nature of the bars and error bars (s.d., s.e.m.)
- If the data are obtained from n {less than or equal to} 2, use scatter plots showing the individual data points.

Discussion of statistical methodology can be reported in the Materials and Methods section, but figure legends should contain a basic description of n, P and the test applied.

See also the guidelines for figure legend preparation:
<https://www.embopress.org/page/journal/14693178/authorguide#figureformat>

10) We now request publication of original source data with the aim of making primary data more accessible and transparent to the reader. Our source data coordinator will contact you to discuss which figure panels we would need source data for and will also provide you with helpful tips on how to upload and organize the files.

11) Our journal encourages inclusion of *data citations in the reference list* to directly cite datasets that were re-used and obtained from public databases. Data citations in the article text are distinct from normal bibliographical citations and should directly link to the database records from which the data can be accessed. In the main text, data citations are formatted as follows: "Data ref: Smith et al, 2001" or "Data ref: NCBI Sequence Read Archive PRJNA342805, 2017". In the Reference list, data citations must be labeled with "[DATASET]". A data reference must provide the database name, accession number/identifiers and a resolvable link to the landing page from which the data can be accessed at the end of the reference. Further instructions are available at .

12) Please also note our reference format:

13) We now use CRediT to specify the contributions of each author in the journal submission system. CRediT replaces the author contribution section, which should be removed from the manuscript. Please use the free text box to provide more detailed descriptions. See also guide to authors:

14) As part of the EMBO publications' Transparent Editorial Process, EMBO reports publishes online a Review Process File to accompany accepted manuscripts. This File will be published in conjunction with your paper and will include the referee reports, your point-by-point response and all pertinent correspondence relating to the manuscript.

You can opt out of this by letting the editorial office know (emboreports@embo.org). If you do opt out, the Review Process File link will point to the following statement: "No Review Process File is available with this article, as the authors have chosen not to make the review process public in this case."

I look forward to seeing a revised version of your manuscript when it is ready. Please let me know if you have any questions or comments regarding the revision.

Yours sincerely,

Ioannis Papaioannou, PhD
Editor
EMBO reports

Referee #1:

In the study "Control of coronary lymphangiogenesis by epicardial VEGF-C/D" de la Cruz and coworkers investigated mechanisms that guide coronary lymph vessel formation in the mouse fetus. They notes that lymph vessels do not replicate the pattern of arteries and veins but grow surface proximal in association with the epicardium. The position of the developing lymphatic plexi between the organ surface and the outermost veins suggested to the authors an important function of the epicardium, which they probed by induced forced expression of VEGF-C under a Wt1-Cre driver or conditional deletion of VEGF-C under the same Wt1-Cre or a Tbx18-Cre driver. Forced expression resulted in extended lymph vessel (LV) coverage, while Wt1-cre-driven deletion caused reduced LV coverage characterized by shorter vessel extension and reduced branching and Tbx18-driven deletion caused a nearly complete failure of coronary LVs to form. Because VEGF-D was found expressed in the epicardium associated with growing LVs, the authors generated a CRISPR ko mouse line, in which exon 3 and 4 of VEGF-D were deleted in order to test a potential cooperation of both factors in coronary LV formation. In agreement with the published literature, deletion of VEGF-D had no effect. Compound mice carrying a full deletion of VEGF-D and a Mt1-cre driven deletion of VEGF-C displayed an aggravated phenotype compared to single VEGF-C deletion, however only in females. Because VEGF-D is an x-linked gene and ko mice are hemizygous the authors speculate about an unsuspected sex-specific role of this cytokine, which is a novel and interesting notion. However, a number of central questions regarding this topic remain open, mainly concerning the biological relevance of this observation and the underlying mechanisms.

Major points:

The authors uses Wt1-cre and Tbx18-cre as independent epicard-specific drivers for Vegfc deletion. Both drivers result in significantly different phenotypes of the lymphatic vasculature, why is that? The authors speculate that Wt1-cre may delete with lower efficiency compared to Tbx18-cre. Does that refer to the temporal onset of activity of Mt1-cre activity or to the extent of converted cells in the tissue? Would a Tbx18-cre;R26-tdTom line show a different labelling pattern compared to the one shown Fig. 1C? Can the efficiency of deletion be assessed by qRT-PCR for VEGF-C on isolated hearts? Alternatively and more elegantly RNA transcript levels could be directly analysed, e.g. by RNA-scope, smFISH, etc.. This point deserves careful investigation, as the residual VEGF-C activity present after deletion with the Mt1-cre driver forms the basis for the reported sex-specific limited redundancy between VEGF-C and D in females. Due to the nearly complete absence of LVs in Tbx1-cre;Vegf fl/fl mice, an analysis of LVs would not be informative

The generation of a new VEGF-D ko line is commendable, but in light of the unexpected sex-related difference in compound-deleted fetuses requires a more precise characterization. Is the RNA transcript encoding the N-terminal domain of pro-VEGF-D (first 106 aa) in the ko line stable, and would the residual N-terminal domain be generated? If produced (either RNA or protein

could develop some spurious activity) it would constitute a difference between females (2 alleles) and males however the differences between females and males then result from the reagent employed. In addition, depending on the epitope recognized by the VEGF-D antibody, the new VEGF-D ko line could be a perfect tool to validate the specificity of the VEGF-D staining in Fig. 3. The pattern suggests that VEGF-D decorates the surface of the producing epicardial cells in what appears to be a polarized fashion, which is a potentially interesting observation.

Minor points:

A variety of microscopes was listed in the methods section, in addition it should be indicated which instrument was used to acquire which data set.

Fig. 2, A, B, E, F - the meaning of the white broken lines should be indicated. Staining for a second positively identifying marker (e.g. Prox1 or VEGFR3) would have been helpful for the interpretation of the wholemount stainings.

Fig. 3. The staining depicted in Fig. 4 C, C' shows a strongly reduced number of LYVE1+ MaP at the base of the heart, is that representative or just coincidence. It so do the macrophages normally express VEGF-D?

Line 204: typo -replace clothing by clotting

Line 212: typo - Slides were thawed at RT...

Referee #2:

The manuscript by de la Cruz et al investigate the contribution of epicardial cells in regulation of cardiac lymphangiogenesis during development in mice.

For this study, the authors generated several elegant mouse models, including two different models of epicardial-selective deletion of the *Vegfc* gene, one model of epicardial-selective *Vegfc* gain of function, and one CRISPR-cas9 model of global *Vegfd* knockout, as well as compound models of epicardial-selective deletion of *Vegfc* gene crossed with *Vegfd* knockout mice. The authors apply careful evaluations of lymphatic structure in 3D images generated at embryonic day 15.5 or 16.5, coupled to assessment of arterial and venous vascular branches, to demonstrate that cardiac lymphatics remain in the epicardial surface and do not follow arteries and veins when these grow into the expanding myocardium during development. Moreover, they demonstrate that lymphatic development is essentially controlled by epicardial-secreted lymphangiogenic growth factors *Vegfc* and *Vegfd*. Interestingly, they demonstrate a more severe phenotype of *Vegfd* knockdown selectively in females but not male embryos. This work agrees with previous studies revealing that cardiac lymphatics are essentially located in the subepicardial layers (Flaht A 2012, cited in the current manuscript), and further that there may be sex-differences in lymphatics. Indeed, circulating *Vegfc* and *Vegfd* levels are elevated in women vs men (Silha JV et al. Int J Obes 2005, not cited in the current manuscript), and this may be partially linked to oestrogen stimulation in females of *Flt4* and *Vegfd* expression (Morfoisse F et al. ATVB 2018, not cited in the current manuscript).

For Fig. 4, please comment on potential sex-differences in lymphatic density in male vs female wt mice and *Wt1Cre x Vegfcfl/fl* parental strain. Please also comment on previous papers that have suggested slightly elevated lymphatic density in female C57Bl/6 mouse healthy hearts (such as Trincot C et al. Circ Res 2019, or Heron et al Circ Res 2022, both not cited in the current manuscript) as compared to male C57Bl/6 mice.

The authors suggest that male 'heterozygous' *Vegfd* genotype may explain why females have more apparent phenotype to *Vegfd* knockdown, as they are 'homozygous' for *Vegfd* due to this gene being located on the X chromosome. However, in females, one X chromosome is rendered inactive. Thus, it seems rather inappropriate to refer to *Vegfd*-deleted females as *Vegfd*^{-/-} and males as *Vegfd*^{-/Φ}. What is the data in favour of higher cardiac *Vegfd* gene or protein levels in female vs male wt mouse hearts?

The authors also conclude that dorsal lymphatics are more resistant to *Vegfd* loss as compared to ventral lymphatics, and suggest that this may reflect their differing embryonic origins. Indeed, a recent study (Martucciello et al FASEB J 2020, not cited in the current manuscript) demonstrated that ventral lymphatics, but not dorsal, are more sensitive to reduced *Flt4* levels. Perhaps non-venous-derived ventral lymphatics express lower *Flt4* levels, which would mean that reduced *Vegfc* or *Vegfd* signaling, as performed in the current manuscript, would be expected to affect preferentially outgrowth of ventral lymphatic? However, the authors report that *Vegfc* knockdown was more potent to reduce dorsal lymphatics (Fig. 2g, h). It would indeed be useful to demonstrate whether protein levels of lymphangiogenic growth factors especially *Vegfc*, *Vegfd*, but perhaps also *Cxcl12* or adrenomedullin, are different in male vs female wt mouse dorsal vs ventral epicardium. Further, the authors conclude that ventral lymphatics in females are more sensitive to *Vegfd* deletion, however the images shown in Fig. 4C and C' vs B and B' do not readily allow for this conclusion. Perhaps tone down the statement about a regional role of *Vegfd* in cardiac lymphatic development.

Previous studies (Liu et al 2021, cited in the current manuscript) have demonstrated that loss of cardiac lymphatics results in developmental cardiac hypoplasia, in part due to reduced *Reelin* levels. It is not apparent from the current images whether cardiac size was affected in *Vegfc* or *Vegfd* deleted mice? Please comment.

Previous work (Makinen et al Nature Med 2001, not cited in the current manuscript) has demonstrated that soluble *Flt4*

transgenic mice, which essentially lack Vegfc and Vegfd signalling, who survive the developmental period go on to establish in the first postnatal weeks some lymphatic vessels in the heart. Thus, with these new elegant mouse models developed by the team of de la Cruz et al it would be very interesting if the authors could evaluate whether epicardial-deleted Vegfc embryos develop cardiac lymphatics postnatally, perhaps linked to macrophage production of lymphangiogenic factors. This experiment may reveal whether: 1) other cell sources, beyond epicardial cells, are sufficient for physiological cardiac lymphangiogenesis in the adult (as demonstrated recently for pathological lymphangiogenesis, Glinton KE JCI 2022); and 2) whether Vegfc and Vegfd have differential roles for cardiac lymphangiogenesis in development versus postnatally.

In all figure legends, please indicate size of scale bars.

Referee #3:

In this article Ester de la Cruz and colleagues propose that coronary lymphatic vessels do not follow the cues provided by blood vessels. This contrasts with lymphatic vessels of the other tissues.

Major concern

1. The idea that the lymphatic vessels always follow blood vessels in other tissues is somewhat misguided. Please see Figure 1 of Liu et al., Cell Reports. 2016, Figure 3A by Matinez-Corral., Circ Res. 2015, Figure 3A by Cha et al., Genes and Dev 2016 and Figure 1 by Baluk et al., Amer J Pathol 2020. While the upstream portions of lymphatic vessels (in the dermis and lungs) do indeed line up closely with blood vessels, the capillaries branch away from the blood vessels. In fact, Liu et al showed that the process of branching away from the blood vessels is regulated by Semaphorin 3G. Therefore, coronary lymphatic vessels do not appear to be unique in this aspect.

2. The authors have also provided evidence that Vegfc produced by Wt1-Cre and Tbx18-cre lineage (epicardial) cells. They have used both gain of function and loss of function approaches to test this possibility. This is an interesting result. Can the authors delete Vegfc using Tie2-Cre and Sma-Cre? If the lymphatic patterning is unaffected in both cases then they can propose an interesting model that blood vessels are not the source of Vegf-c that regulates lymphatic vessel patterning.

Minor concern

1. The figures are too small and the resolution is low to appreciate many details.
2. The Tbx18-Cre;Vegfc^{f/f} hearts appear abnormally small. Have the authors analyzed these samples carefully?

Referee #1:

In the study "Control of coronary lymphangiogenesis by epicardial VEGF-C/D" de la Cruz and coworkers investigated mechanisms that guide coronary lymph vessel formation in the mouse fetus. They notes that lymph vessels do not replicate the pattern of arteries and veins but grow surface proximal in association with the epicardium. The position of the developing lymphatic plexi between the organ surface and the outermost veins suggested to the authors an important function of the epicardium, which they probed by induced forced expression of VEGF-C under a Wt1-Cre driver or conditional deletion of VEGF-C under the same Wt1-Cre or a Tbx18-Cre driver. Forced expression resulted in extended lymph vessel (LV) coverage, while Wt1-cre-driven deletion caused reduced LV coverage characterized by shorter vessel extension and reduced branching and Tbx18-driven deletion caused a nearly complete failure of coronary LVs to form.

Because VEGF-D was found expressed in the epicardium associated with growing LVs, the authors generated a CRISPR ko mouse line, in which exon 3 and 4 of VEGF-D were deleted in order to test a potential cooperation of both factors in coronary LV formation. In agreement with the published literature, deletion of VEGF-D had no effect. Compound mice carrying a full deletion of VEGF-D and a Mt1-cre driven deletion of VEGF-C displayed an aggravated phenotype compared to single VEGF-C deletion, however only in females. Because VEGF-D is an x-linked gene and ko mice are hemizygous the authors speculate about an unsuspected sex-specific role of this cytokine, which is a novel and interesting notion. However, a number of central questions regarding this topic remain open, mainly concerning the biological relevance of this observation and the underlying mechanisms.

Major points:

The authors uses Wt1-cre and Tbx18-cre as independent epicard-specific drivers for Vegfc deletion. Both drivers result in significantly different phenotypes of the lymphatic vasculature, why is that? The authors speculate that Wt1-cre may delete with lower efficiency compared to Tbx18-cre. Does that refer to the temporal onset of activity of Mt1-cre activity or to the extent of converted cells in the tissue? Would a Tbx18-cre;R26-tdTom line show a different labelling pattern compared to the one shown Fig. 1C'? Can the efficiency of deletion be assessed by qRT-PCR for VEGF-C on isolated hearts? Alternatively and more elegantly RNA transcript levels could be directly analysed, e.g. by RNA-scope, smFISH, etc.. This point deserves careful investigation, as the residual VEGF-C activity present after deletion with the Mt-1-cre driver forms the basis for the reported sex-specific limited redundancy between VEGF-C and D in females. Due to the nearly complete absence of LVs in Tbx1-cre;Vegf fl/fl mice, an analysis of LVs would not be informative.

This is an important point raised by the reviewer. Unfortunately, we have not been able to detect bona fide VEGFC by immunodetection in either whole-mounts or sections. In addition, RNA-scope is not useful to distinguish the loss of exon 3 from the VEGFC transcript, which only spans 191 bp of a ~2kb transcript. While several previous studies show that epicardial recombination of both Tbx18Cre and WT1Cre lines in the epicardium and epicardially derived cells is similar at mid-gestation (Cai et al. Nature (2008) 454, 104–108. 10.1038/nature06969; Christoffels, et al. Nature (2009) 458, E8–E9. 10.1038/nature07916; Wessels et al., Dev Biol. 2012 366:111–24. 10.1016/j.ydbio.2012.04.020; Villa Del Campo et al., Sci Rep. (2016) 18:35366. 10.1038/srep35366), both lines independently recombine a non-overlapping subset of cardiomyocytes. Given the results reported here, it is unlikely that the recombination in cardiomyocytes is relevant in this context, first, because cardiomyocytes express little VEGFC at mid-gestation (Chen et al., Development. 2014 141:4500–12. 10.1242/dev.113639.) and second, because lymphatics do not grow into the myocardium during gestation, as it would be expected if cardiomyocytes were a VEGFC source during this period. The main difference between the two Cre lines used is actually the timing of activation. While Tbx18 is present in the splanchnic mesoderm of the juxta-cardiac region, from which the proepicardium derives (Tyser et al 2021 Mar 5;371(6533):eabb2986. 10.1126/science.abb2986) and therefore the proepicardial organ is already fully recombined (Cai et al. Nature (2008) 454, 104–108. 10.1038/nature06969), the WT1Cre line that we used starts to recombine epicardial cells when they are already on the ventricular surface (Wessels et al., Dev Biol. 2012 366:111–24. 10.1016/j.ydbio.2012.04.020). These differences between these two Cre lines suggest that timely elimination of VEGFC is essential for obtaining a timely elimination of epicardial VEGFC activity. We have now discussed these aspects in the manuscript.

Furthermore, given the difficulties in determining VEGFC expression in situ, we then decided to perform RNAseq of epicardium+subepicardium from Wt1Cre;Vegfc^{fl/fl} and Tbx18 Cre;Vegfc^{fl/fl} fetal hearts. The results show that the Cre cDNA is expressed at higher levels in Tbx18Cre than in WT1Cre hearts, which most likely affects the timing of complete elimination of VEGFC third exon. From the same data we concluded that the elimination of VEGFC third exon is similar between WT1Cre-recombined and Tbx18Cre-recombined hearts at E16.5. This again suggests that differences in timing of complete VEGFC elimination underlie the observed results.

The generation of a new VEGF-D ko line is commendable, but in light of the unexpected sex-related difference in compound-deleted fetuses requires a more precise characterization. Is the RNA transcript encoding the N-terminal domain of pro-VEGF-D (first 106 aa) in the ko line stable, and would the residual N-terminal domain be generated? If produced (either RNA or protein could develop some spurious activity) it would constitute a

difference between females (2 alleles) and males however the differences between females and males then result from the reagent employed.

We have also compared the abundance of VEGFD mRNA in RNAseq comparing heterozygous and homozygous mutant female hearts. While we observed a reduction in VEGFD transcripts, indeed the mutants showed some level of expression. Indeed, this mutant may thus express a peptide encoding the first 106 aas of the VEGFD protein. Regarding the possibility that this peptide is expressed differentially in males and females, dosage compensation by random inactivation of one of the X-chromosomes in females renders X-linked genes equally expressed in males and females.

In addition, depending on the epitope recognized by the VEGF-D antibody, the new VEGF-D ko line could be a perfect tool to validate the specificity of the VEGF-D staining in Fig. 3. The pattern suggests that VEGF-D decorates the surface of the producing epicardial cells in what appears to be a polarized fashion, which is a potentially interesting observation.

The epitope recognized by the VEGFD antibody used –which is the only one we could make work from several tested– only contains 12 aas spared by the mutation introduced (epitope: aa 94-210 ; spared region: aa 1-105). Even though some cross-reaction could take place due to the residual 12 aas, we saw a clear depletion of the staining with this antibody in the VEGFD mutant hearts (results now included in Fig EV2). As mentioned by the reviewer, apparent polarized expression might be present in epicardial cells, however, this is an aspect that has not been confirmed in histological sections and would need a much more thorough characterization.

Minor points:

A variety of microscopes was listed in the methods section, in addition it should be indicated which instrument was used to acquire which data set.

We reviewed this information and in reality, only two microscopes were used. This is now corrected in the manuscript and the images taken with each microscope have been identified

Fig. 2, A, B, E, F - the meaning of the white broken lines should be indicated.

They indicate the limits between the sinus venosus and the ventricles at the dorsal side and between the Pulmonary Artery and Aorta at the ventral side. This has now been indicated in the Figure legend

Staining for a second positively identifying marker (e.g. Prox1 or VEGFR3) would have been helpful for the interpretation of the wholemount stainings.

We now show Lyve1 + Prox1 in these panels

Fig. 3. The staining depicted in Fig. 4 C, C' shows a strongly reduced number of LYVE1+ MaP at the base of the heart, is that representative or just coincidence. It so do the macrophages normally express VEGF-D?

Lyve1+ macrophages are not depleted in the mutants, but depending on the specific confocal stack they may not appear in some areas of the images. From our antibody staining, we do not find VEGFD expression in these macrophages, however, other reports indicate that they express VEGFC. Nonetheless, neither *Wt1^{Cre}* nor *Tbx18^{Cre}* recombine this population, indicating that their VEGFC/D expression is not sufficient to drive lymphangiogenesis.

Line 204: typo -replace clothing by clotting

Corrected, Thank you

Line 212: typo - Slides were thawed at RT...

Corrected, Thank you

Referee #2:

The manuscript by de la Cruz et al investigate the contribution of epicardial cells in regulation of cardiac lymphangiogenesis during development in mice.

For this study, the authors generated several elegant mouse models, including two different models of epicardial-

selective deletion of the *Vegfc* gene, one model of epicardial-selective *Vegfc* gain of function, and one CRISPR-cas9 model of global *Vegfd* knockout, as well as compound models of epicardial-selective deletion of *Vegfc* gene crossed with *Vegfd* knockout mice. The authors apply careful evaluations of lymphatic structure in 3D images generated at embryonic day 15.5 or 16.5, coupled to assessment of arterial and venous vascular branches, to demonstrate that cardiac lymphatics remain in the epicardial surface and do not follow arteries and veins when these grow into the expanding myocardium during development. Moreover, they demonstrate that lymphatic development is essentially controlled by epicardial-secreted lymphangiogenic growth factors *Vegfc* and *Vegfd*. Interestingly, they demonstrate a more severe phenotype of *Vegfd* knockdown selectively in females but not male embryos. This work agrees with previous studies revealing that cardiac lymphatics are essentially located in the subepicardial layers (Flaht A 2012, cited in the current manuscript), and further that there may be sex-differences in lymphatics. Indeed, circulating *Vegfc* and *Vegfd* levels are elevated in women vs men (Silha JV et al. *Int J Obes* 2005, not cited in the current manuscript), and this may be partially linked to oestrogen stimulation in females of *Flt4* and *Vegfd* expression (Morfoisse F et al. *ATVB* 2018, not cited in the current manuscript).

For Fig. 4, please comment on potential sex-differences in lymphatic density in male vs female wt mice and *Wt1Cre x Vegfcfl/fl* parental strain. Please also comment on previous papers that have suggested slightly elevated lymphatic density in female C57Bl/6 mouse healthy hearts (such as Trincot C et al. *Circ Res* 2019, or Heron et al *Circ Res* 2022, both not cited in the current manuscript) as compared to male C57Bl/6 mice.

We thank the reviewer for pointing this out. We have now compared lymphatic vessel density between male and female control hearts of the genetic background of our double VEGFC/D experiments. These measurements were made on E16.5 hearts and show a strong tendency to higher lymphatic vessel coverage in females as compared to males. This indicates that the observations of Trincot et al. in adult mice starts to build up during development (In Heron et al., 2022, we have not found any reference to sex-specific differences in coronary lymphatic vasculature). One possibility is that the higher lymphatic vessel density in females might be promoted by VEGFD, given the specific sensitivity of females to VEGFD loss. The reason why VEGFD affects females and not males, however, remains to be determined. We now comment these results and the relevant literature in the manuscript and included the sex-specific study in Supplementary Figure 2.

The authors suggest that male 'heterozygous' *Vegfd* genotype may explain why females have more apparent phenotype to *Vegfd* knockdown, as they are 'homozygous' for *Vegfd* due to this gene being located on the X chromosome. However, in females, one X chromosome is rendered inactive. Thus, it seems rather inappropriate to refer to *Vegfd*-deleted females as *Vegfd*^{-/-} and males as *Vegfd*^{-/Φ}. What is the data in favour of higher cardiac *Vegfd* gene or protein levels in female vs male wt mouse hearts?

We may have not explained ourselves correctly, because we did not mean what the reviewer expresses. Given that VEGFD is on the X chromosome, females can be wild type, heterozygous or homozygous for VEGFD mutations. In contrast, males can only be wild type or hemizygous. Hemizygoty refers to a condition in which only one allele of a gene is present. In this case, males hemizygous for a knockout allele are null-hemizygous for VEGFD and the equivalent situation in females is homozygosity for the same knockout allele. In summary, in our experiments, we compare VEGFD-null females to VEGFD-null males. Under these conditions, the fact that females inactivate one of the X chromosomes is irrelevant. It would be relevant if we analyzed the heterozygous conditions in females, but this is not the case (precisely, because such experiments would entail uncontrolled randomness). Representation of hemizygous alleles in males as *Vegfd*^{-/Φ} is according to standard, however if the editorial policy is different, we would be happy to modify this. There is no data indicating higher VEGFD in females versus males and, as explained above to reviewer 1, dosage compensation leads to equal expression levels between males and females for X-linked genes.

The authors also conclude that dorsal lymphatics are more resistant to *Vegfd* loss as compared to ventral lymphatics, and suggest that this may reflect their differing embryonic origins. Indeed, a recent study (Martucciello et al *Faseb J* 2020, not cited in the current manuscript) demonstrated that ventral lymphatics, but not dorsal, are more sensitive to reduced *Flt4* levels. Perhaps non-venous-derived ventral lymphatics express lower *Flt4* levels, which would mean that reduced *Vegfc* or *Vegfd* signaling, as performed in the current manuscript, would be expected to affect preferentially outgrowth of ventral lymphatic?

We thank the reviewer for this suggestion. We have now quantified VEGFR3 expression by immunofluorescence in dorsal versus ventral lymphatics and found no differences (Data included in Supplementary Figure 2), which indicates that this may not be a determinant of the different sensitivity of the dorsal and ventral sides to VEGFC elimination.

Martucciello et al report the ability of VEGFR3 overexpression to rescue the defects in the lymphatic vasculature in *Tbx1*^{+/-};VEGFR3^{+/-} hearts only in the dorsal but not in the ventral side. The interpretation of these data is difficult, given the dual role of *Tbx1* in ventral lymphatic vessels. *Tbx1* regulates the addition of the second heart field precursors to the heart and, as demonstrated by us and others, the ventral lymphatics show an essential

contribution from the SHF precursors (Lioux et al Dev Cell 2020; Maruyama et al Dev Biol 2019) and ventral lymphatics do not develop in the absence of Tbx1 (Lioux et al Dev Cell 2020).

However, the authors report that Vegfc knockdown was more potent to reduce dorsal lymphatics (Fig. 2g, h). It would indeed be useful to demonstrate whether protein levels of lymphangiogenic growth factors especially Vegfc, Vegfd, but perhaps also Cxcl12 or adrenomedullin, are different in male vs female wt mouse dorsal vs ventral epicardium.

We tried several antibodies for detection in immunofluorescence and could reliably obtain results from Cxcl12, and VEGFD but not for others. Whereas we did not find differences in VEGFD expression, we found a mild (25%) higher expression of Cxcl12 on the dorsal side. These data are now presented in Figure EV1 together with similar data from VEGFR3 detection. The question remains whether this difference in Cxcl12 expression is functionally relevant. To more broadly study this aspect, we adopted an unbiased approach performing RNAseq on isolated dorsal and ventral epicardia/subepicardia and examined the expression of mRNA encoding lymphangiogenic factors. We found that 8 out of 22 examined lymphangiogenic molecules were expressed at higher levels on the dorsal side. While not significant, Cxcl12 mRNA showed higher expression on the dorsal side, in correlation with the results obtained in immunofluorescence. These results suggest a generally more angiogenic environment on the ventral side of the heart at this stage but also point out to exceptions to this rule for some factors, like Cxcl12. These results are now included in Figure EV1.

To then study differences between males and females, we performed similar RNAseq experiments comparing both sexes. In this case, we did not find any differences in the expression of lymphangiogenic molecules and therefore the basis for the differential coronary lymphangiogenesis between males and females remains unknown. These results are reported in Figure EV3.

Further, the authors conclude that ventral lymphatics in females are more sensitive to Vegfd deletion, however the images shown in Fig. 4C and C' vs B and B' do not readily allow for this conclusion. Perhaps tone down the statement about a regional role of Vegfd in cardiac lymphatic development.

The reviewer is correct. The evidence on this point is statistically significant but weak. We have therefore eliminated these comments

Previous studies (Liu et al 2021, cited in the current manuscript) have demonstrated that loss of cardiac lymphatics results in developmental cardiac hypoplasia, in part due to reduced Reelin levels. It is not apparent from the current images whether cardiac size was affected in Vegfc or Vegfd deleted mice? Please comment.

We have now measured the heart-to-body weight in VEGFC/VEGFD double mutant male hearts at postnatal day 20 and found that indeed these hearts are smaller and are so to an extent similar to that previously described by Liu et al. Nature 2021. These data are presented in supplementary Figure 1A.

Previous work (Makinen et al Nature Med 2001, not cited in the current manuscript) has demonstrated that soluble Flt4 transgenic mice, which essentially lack Vegfc and Vegfd signalling, who survive the developmental period go on to establish in the first postnatal weeks some lymphatic vessels in the heart. Thus, with these new elegant mouse models developed by the team of de la Cruz et al it would be very interesting if the authors could evaluate whether epicardial-deleted Vegfc embryos develop cardiac lymphatics postnatally, perhaps linked to macrophage production of lymphangiogenic factors. This experiment may reveal whether: 1) other cell sources, beyond epicardial cells, are sufficient for physiological cardiac lymphangiogenesis in the adult (as demonstrated recently for pathological lymphangiogenesis, Glington KE JCI 2022); and 2) whether Vegfc and Vegfd have differential roles for cardiac lymphangiogenesis in development versus postnatally.

We thank the reviewer for this suggestion. Animals in which we eliminated VEGFC with WT1Cre and are simultaneously mutant for VEGFD survive postnatally, and we now used these animals to determine the presence and abundance of cardiac lymphatic vessels in adult female mice. The results (shown in Figure 4G-I) indicate partial recovery of the dorsal lymphatics but a remaining strong affection of the ventral ones. These data indicate the presence of alternative postnatal lymphangiogenic pathways that mainly act on the dorsal side of the heart but also a persistent requirement for epicardial VEGFC/D in postnatal lymphangiogenesis.

In all figure legends, please indicate size of scale bars.

Done

Referee #3:

In this article Ester de la Cruz and colleagues propose that coronary lymphatic vessels do not follow the cues provided by blood vessels. This contrasts with lymphatic vessels of the other tissues.

Major concern

1. The idea that the lymphatic vessels always follow blood vessels in other tissues is somewhat misguided. Please see Figure 1 of Liu et al., *Cell Reports*. 2016, Figure 3A by Matinez-Corral., *Circ Res*. 2015, Figure 3A by Cha et al., *Genes and Dev* 2016 and Figure 1 by Baluk et al., *Amer J Pathol* 2020. While the upstream portions of lymphatic vessels (in the dermis and lungs) do indeed line up closely with blood vessels, the capillaries branch away from the blood vessels. In fact, Liu et al showed that the process of branching away from the blood vessels is regulated by Semaphorin 3G. Therefore, coronary lymphatic vessels do not appear to be unique in this aspect.

We than the reviewer for these comments. The statement that coronary lymphatic vessels grow following coronary blood vessels is found in several review articles and text books. We think it is therefore an important aspect to highlight, especially given that our study characterizes the development of both collectors and capillaries. We appreciate that a similar situation might take place in other tissues and we do not claim that this is exclusive of coronary lymphatic vessels. We have nonetheless now referred specifically to collector vessels and replaced “several tissues” by “some tissues”, when we refer to the fact that lymphatics follow blood vessels as they grow.

2. The authors have also provided evidence that Vegfc produced by Wt1-Cre and Tbx18-cre lineage (epicardial) cells. They have used both gain of function and loss of function approaches to test this possibility. This is an interesting result. Can the authors delete Vegfc using Tie2-Cre and Sma-Cre? If the lymphatic patterning is unaffected in both cases then they can propose an interesting model that blood vessels are not the source of Vegf-c that regulates lymphatic vessel patterning.

We thank the reviewer for this suggestion, however, our goal in this manuscript is to address the role of the epicardium in coronary lymphangiogenesis and generating new complex compound mutants would represent a significant amount of time and effort beyond a standard manuscript revision. While we cannot exclude a role of coronary vessel endothelium or smooth muscle in coronary lymphangiogenesis, we think our data demonstrate a major role for epicardial VEGFC and D in coronary lymphangiogenesis. The contribution of the WT1-Cre lineage to coronary endothelial cells is reported to be very minor (Zhou et al., *Nature*. 2008; 454:109–113) and that of Tbx18-Cre cells is absent (Cai et al *Nature*. 2008;454:104–108; Grieskamp et al, *Circulation Research* 10.1161/CIRCRESAHA.110.228809) and therefore our result using both lines independently rule out that blood endothelial cells play a major role in coronary lymphangiogenesis (refs). Regarding the smooth muscle, while the epicardial contribution to vascular coronary smooth muscle is important, the contribution of coronary vascular smooth muscle to VEGFC production is minor compared to that of fibroblasts or epicardium, as can be seen in single-cell transcriptomic studies (see data in Feng et al., *Nat. Comm.* <https://doi.org/10.1038/s41467-022-35691-Z>, for example). Below, we reproduce the distribution of VEGFC expression in different cell populations of the heart during gestation in two different strains of mice (data from Feng et al *Nat. Comm.* retrieved from: https://cells-test.gi.ucsc.edu/?ds=mouse-dev-heart_smooth). In addition, smooth muscle colonizing the base of the ventricles (the regions close to the great arteries) mostly derives from neural crest and second heart field, which are not recombined by either Tbx18Cre or WT1Cre (Arima et al., *Nature Communications* 2012; 10.1038/ncomms2258). Our results show complete agenesis of ventricular lymphatics –including those at the base of the ventricles– in epicardial VEGFC mutants. We therefore think that the results presented support the conclusion that the predominant lymphangiogenic signals in the heart are directly produced by the epicardium and subepicardial-derived fibroblasts, but not coronary blood vessels. We have now added this important discussion to the manuscript.

Figure from Feng et al Nat. Comm. not shown.

Minor concern

1. The figures are too small and the resolution is low to appreciate many details.

We now provided larger images

2. The *Tbx18-Cre;Vegf^{cf/f}* hearts appear abnormally small. Have the authors analyzed these samples carefully?

We did not directly determined the size of the *Tbx18-Cre;VEGFC^{ff}* hearts, but we now provide data showing that even in the milder condition of deletion with *WT1^{Cre}*, the hearts are smaller, which is in agreement with the previously published evidence of smaller hearts when the lymphatic vasculature is compromised (Liu et al Nature 2020)

Dear Dr. Torres,

Thank you for submitting your revised manuscript. It has now been seen by two of the original referees. I apologize for this unusual delay in getting back to you, it took longer than anticipated to receive the referee reports due to this busy time of the year.

As you can see, both referees find that the study is significantly improved during revision and recommend publication. However, I need you to address the points below before I can accept the manuscript.

- Please address the remaining minor concerns of referee #1 and provide a point-by-point response.
- Please rename the 'Results' section as 'Results and discussion'.
- Please provide 3-5 keywords for your study. These will be visible in the html version of the paper and on PubMed and will help increase the discoverability of your work.
- Please add a link that directly resolves to the dataset GSE284800 into the Data Availability section. Moreover, please make GSE284800 publicly available and remove the reviewer token. Please move the following sentence to the Methods section: "Mouse strains described here are available upon request."
- Please rename the 'Declaration of Interests' section as 'Disclosure Statement & Competing Interests'.
- Please remove the 'Author Contributions' section from the manuscript text.
- Please make sure that all acknowledged funders in the manuscript are also entered into the manuscript tracking system - e.g. EMBO Short-Term Fellowship number 8357 is currently missing in the manuscript tracking system.
- All research articles submitted as revised versions must include a structured methods section that includes a Reagents and Tools Table followed by a Methods and Protocols section. Please see <https://www.embopress.org/page/journal/14693178/authorguide#structuredmethods> for further information.
- During our routine figure checks, we note a potential image re-use between Figure 3A & Figure EV2A. Please clarify, also in the respective figure legends of both panels.
- Our production/data editors have asked you to clarify several points in the figure legends:
 - o Please note that the exact p values are not provided in the legends of figures 4E, F, I; EV1 C,
 - o Please indicate the statistical test used for data analysis in the legends of figures EV1 A-C, J; EV2 B"
 - o Please note that information related to n is missing in the legends of figures 2C, D, F, G, H, I, J, K, N, O; 3F-K; 4E, F, I; EV1 K; EV2 B.
 - o Please note that the error bars are not defined in the legends of figures 2C, D, F, G, H, I, J, K, N, O; 3F-K; 4E, F, I; EV1 A, B, C, E, G, I, J."
- Papers published in EMBO Reports include a 'synopsis' and 'bullet points' to further enhance discoverability. Both are displayed on the html version of the paper and are freely accessible to all readers. The synopsis includes a short standfirst summarizing the study in 1 or 2 sentences (max 35 words) that summarize the paper and are provided by the authors and streamlined by the handling editor. I would therefore ask you to include your synopsis blurb and 3-5 bullet points listing the key experimental findings.
- In addition, please provide an image for the synopsis. This image should provide a rapid overview of the question addressed in the study but still needs to be kept fairly modest since the image size cannot exceed 550 (width) x 300-600 (height) pixels.

Thank you again for giving us to consider your manuscript for EMBO Reports, I look forward to your minor revision.

Kind regards,

Deniz Senyilmaz Tiebe

--

Deniz Senyilmaz Tiebe, PhD
Senior Scientific Editor
EMBO Reports

Referee #1:

Here, de la Cruz and coworkers present a revised version of their manuscript "Control of coronary lymphangiogenesis by epicardial VEGF-C/D".

A major point raised during the review referred to the difference in coronary lymph vessel formation in two different new VEGF-C deletion models, where either Tbx18-Cre or Wt1-Cre deleter lines were used to delete VEGF-C in epicardium and epicard-derived cells.

The authors explain that in addition to the reported earlier onset of Tbx-18 expression, they detect in RNA-seq data a moderately higher expression of Cre-recombinase in Tbx-18 compared to Wt1-Cre -mice, resulting in a significantly earlier completion of VEGF-C depletion. The developmentally earlier complete deletion satisfactorily explains the observed differences.

New RNA-seq data in the revised manuscript suggest a more pro-lymphangiogenic environment of the ventral subepicardium with the exception of CXCL12, which was the single pro-lymphangiogenic factor found more strongly expressed on the dorsal side of the heart. The dorsal side had shown increased sensitivity to Wt1-Cre driven VEGF-C loss. The authors discuss that neither expression of VEGF-D nor its receptor VEGFR-3 displayed dorso-ventral differences, which they attribute to a generally more lymphangiogenic ventral environment. In view of the "...strong expression of the lymphatic endothelial cell guidance cytokine CXCL12 in the epicardium, together with VEGFC and VEGFD ..." (line 194-196), it might be interesting to discuss a recent study (Development 151(22): dev202947), which reports enhanced VEGFR-3 surface expression in the presence of CXCR4/CXCL12 signaling.

Minor points:

Lines 7 - 12 list three affiliations, however, the third affiliation is not associated with an author

Line 75 refers to E14.5, however, the corresponding data presented in Fig. 1 (A-B') are derived from E16.5 fetuses.

Line 113, start sentence with capital letter

Line 122, words seems missing - difference?

Line 157, what are the values 82% and 75% reduction referring to? Is this the %age reduction of compound ko compared to WT? This seems unlikely, in contrast to the statement in the text, the effect of combined VEGF-C/D loss appears to be more severe on the dorsal side (Fig 4 E+F).

Line 287, add information on the Cxcl12 antibody to the materials and Methods section.

Referee #2:

The revision perfectly addressed all the questions. No more comments or questions. It's a pleasure to read!

- Please rename the 'Results' section as 'Results and discussion'.
Done
- Please provide 3-5 keywords for your study. These will be visible in the html version of the paper and on PubMed and will help increase the discoverability of your work.
Done
- Please add a link that directly resolves to the dataset GSE284800 into the Data Availability section. Moreover, please make GSE284800 publicly available and remove the reviewer token. Please move the following sentence to the Methods section: "Mouse strains described here are available upon request."
Done
- Please rename the 'Declaration of Interests' section as 'Disclosure Statement & Competing Interests'.
Done
- Please remove the 'Author Contributions' section from the manuscript text.
Done
- Please make sure that all acknowledged funders in the manuscript are also entered into the manuscript tracking system - e.g. EMBO Short-Term Fellowship number 8357 is currently missing in the manuscript tracking system.
Done

- All research articles submitted as revised versions must include a structured methods section that includes a Reagents and Tools Table followed by a Methods and Protocols section. Please see <https://www.embopress.org/page/journal/14693178/authorguide#structuredmethods> for further information.
Done

- During our routine figure checks, we note a potential image re-use between Figure 3A & Figure EV2A. Please clarify, also in the respective figure legends of both panels.
This is due to the fact that Figure 3A is shown for qualitative appreciation of the CXCL12 expression and the specimen forms part of the collection of hearts quantified in EV2A. We have now explained this circumstance in both Figure legends

- Our production/data editors have asked you to clarify several points in the figure legends:
 - o Please note that the exact p values are not provided in the legends of figures 4E, F, I; EV1 C, In these cases, the program used for the statistic (PRISM) only provided the information that the p-values are smaller than 0.0001 but did not yield an exact p-value; therefore the exact p-value is not available to us but obviously very far away from non-significance
 - o Please indicate the statistical test used for data analysis in the legends of figures EV1 A-C, J; EV2 B"
Done
 - o Please note that information related to n is missing in the legends of figures 2C, D, F, G, H, I, J, K, N, O; 3F-K; 4E, F, I; EV1 K; EV2 B.
Included
 - o Please note that the error bars are not defined in the legends of figures 2C, D, F, G, H, I, J, K, N, O; 3F-K; 4E, F, I; EV1 A, B, C, E, G, I, J."
Done and included some error bars that were missing

- Papers published in EMBO Reports include a 'synopsis' and 'bullet points' to further enhance discoverability. Both are displayed on the html version of the paper and are freely accessible to all readers. The synopsis includes a short standfirst summarizing the study in 1 or 2 sentences (max 35 words) that summarize the paper and are provided by the authors and streamlined by the handling editor. I would therefore ask you to include your synopsis blurb and 3-5 bullet points listing the key experimental findings.
Done
- In addition, please provide an image for the synopsis. This image should provide a rapid overview of the question addressed in the study but still needs to be kept fairly modest since the image size cannot exceed 550 (width) x 300-600 (height) pixels.
Done

Referee #1:

Here, de la Cruz and coworkers present a revised version of their manuscript "Control of coronary lymphangiogenesis by epicardial VEGF-C/D".

A major point raised during the review referred to the difference in coronary lymph vessel formation in two different new VEGF-C deletion models, where either Tbx18-Cre or Wt1-Cre deleter lines were used to delete VEGF-C in epicardium and epicard-derived cells.

The authors explain that in addition to the reported earlier onset of Tbx-18 expression, they detect in RNA-seq data a moderately higher expression of Cre-recombinase in Tbx-18 compared to Wt1-Cre -mice, resulting in a significantly earlier completion of VEGF-C depletion. The developmentally earlier complete deletion satisfactorily explains the observed differences.

New RNA-seq data in the revised manuscript suggest a more pro-lymphangiogenic environment of the ventral subepicardium with the exception of CXCL12, which was the single pro-lymphangiogenic factor found more strongly expressed on the dorsal side of the heart. The dorsal side had shown increased sensitivity to Wt1-Cre driven VEGF-C loss. The authors discuss that neither expression of VEGF-D nor its receptor VEGFR-3 displayed dorso-ventral differences, which they attribute to a generally more lymphangiogenic ventral environment. In view of the "...strong expression of the lymphatic endothelial cell guidance cytokine CXCL12 in the epicardium, together with VEGFC and VEGFD ..." (line 194-196), it might be interesting to discuss a recent study (Development 151(22): dev202947), which reports enhanced VEGFR-3 surface expression in the presence of CXCR4/CXCL12 signaling.

We thank the reviewer for this suggestion, which we incorporated to the manuscript. We believe the intended reference is dev202901, not dev202947, which does not deal with CXCL12/CXCR4.

Minor points:

Lines 7 - 12 list three affiliations, however, the third affiliation is not associated with an author

corrected

Line 75 refers to E14.5, however, the corresponding data presented in Fig. 1 (A-B') are derived from E16.5 fetuses.

corrected

Line 113, start sentence with capital letter

corrected

Line 122, words seems missing - difference?

corrected

Line 157, what are the values 82% and 75% reduction referring to? Is this the %age reduction of compound ko compared to WT? This seems unlikely, in contrast to the statement in the text, the effect of combined VEGF-C/D loss appears to be more severe on the dorsal side (Fig 4 E+F).

In fact the percentages were swapped. We now refer only to the significance of the results

Line 287, add information on the Cxcl12 antibody to the materials and Methods section.

Included

Dr. Miguel Torres
Centro Nacional de Investigaciones Cardiovasculares
Desarrollo y Reparación Cardiovascular
3, Melchor Fernández Almagro
Madrid, Madrid 28029
Spain

Dear Dr. Torres,

Thank you for submitting your revised manuscript. I have now looked at everything and all is fine. Therefore, I am very pleased to accept your manuscript for publication in EMBO Reports.

Congratulations on a nice work!

Before we can transfer your manuscript to our production team, I need your input on one more point. I took the liberty of suggesting a minor change in the title to increase clarity and accessibility of the findings. Please take a look and confirm, or feel free to propose further changes. Thank you.

"Epicardial VEGFC/D signaling is essential for coronary lymphangiogenesis"

Kind regards,

Deniz Senyilmaz Tiebe

--

Deniz Senyilmaz Tiebe, PhD
Senior Scientific Editor
EMBO Reports
